# Long-tailed Adversarial Training with Self-Distillation

**Seungju Cho\*, Hongsin Lee\*, Changick Kim**
Korea Advanced Institute of Science and Technology (KAIST)
{joyga, hongsin04, changick}@kaist.ac.kr

## Abstract

Adversarial training significantly enhances adversarial robustness, yet superior performance is predominantly achieved on balanced datasets. Addressing adversarial robustness in the context of unbalanced or long-tailed distributions is considerably more challenging, mainly due to the scarcity of tail data instances. Previous research on adversarial robustness within long-tailed distributions has primarily focused on combining traditional long-tailed natural training with existing adversarial robustness methods. In this study, we provide an in-depth analysis for the challenge that adversarial training struggles to achieve high performance on tail classes in long-tailed distributions. Furthermore, we propose a simple yet effective solution to advance adversarial robustness on long-tailed distributions through a novel self-distillation technique. Specifically, this approach leverages a balanced self-teacher model, which is trained using a balanced dataset sampled from the original long-tailed dataset. Our extensive experiments demonstrate state-of-the-art performance in both clean and robust accuracy for long-tailed adversarial robustness, with significant improvements in tail class performance on various datasets. We improve the accuracy against PGD attacks for tail classes by 20.3, 7.1, and 3.8 percentage points on CIFAR-10, CIFAR-100, and Tiny-ImageNet, respectively, while achieving the highest robust accuracy.

## 1 Introduction

Recent studies have highlighted the vulnerabilities inherent in deep learning models when subjected to adversarial attacks (Goodfellow et al., 2014; Carlini & Wagner, 2017; Madry et al., 2017; Athalye et al., 2018). These attacks exploit subtle changes in input data that can lead to drastically incorrect predictions, undermining model reliability in critical applications (Ma et al., 2021; Grigorescu et al., 2020; Wang et al., 2023). As a result, research efforts have focused on enhancing robustness against such adversarial threats, with various strategies being explored (Das et al., 2017; Xie et al., 2019; Cohen et al., 2019; Carmon et al., 2019; Zhang et al., 2022; Jin et al., 2023). Among these, adversarial training (Goodfellow et al., 2014; Madry et al., 2017) has proven to be one of the most effective methods for enhancing model robustness (Pang et al., 2020; Bai et al., 2021a; Wei et al., 2023). However, many existing studies primarily validate their approaches on balanced datasets, overlooking the practical scenarios where data is inherently imbalanced or long-tailed. This gap underscores the need for novel adversarial training methodologies capable of addressing these more complex data distributions effectively.

While numerous studies (Cao et al., 2019a; Cui et al., 2019; Kang et al., 2019; Zhou et al., 2020; Li et al., 2021; Alshammari et al., 2022; Du et al., 2023) have addressed long-tailed distributions without considering robustness, the intersection of adversarial training and long-tailed distributions (Wu et al., 2021; Li et al., 2023; Yue et al., 2024) has received far less attention. Existing research in this area primarily combines traditional long-tailed classification techniques with basic adversarial training methods, such as PGD adversarial training (Madry et al., 2017) and TRADES loss (Zhang et al., 2019) with balanced softmax (Wu et al., 2021; Yue et al., 2024). Despite combining such methods, existing approaches still demonstrate low performance on tail classes with fewer samples in long-tailed distributions. We find that their high robustness primarily stems from the improved

---

*These authors contributed equally.

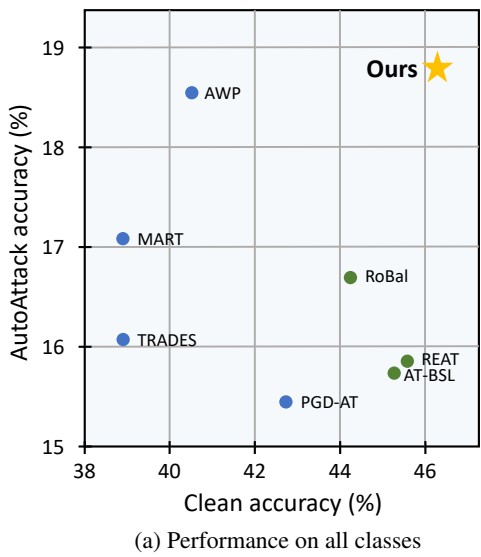 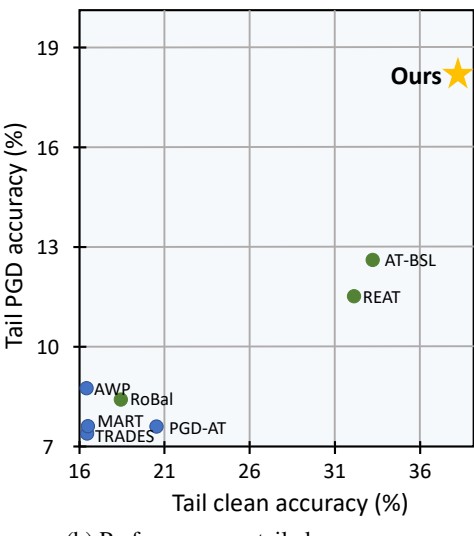

(a) Performance on all classes        (b) Performance on tail classes

Figure 1: **(a)** The overall clean accuracy and AutoAttack (Croce & Hein, 2020) accuracy of various adversarial training methods (*blue circles*) and long-tailed adversarial training methods (*green circles*) using the ResNet-18 (He et al., 2016a) architecture on CIFAR-100-LT (Krizhevsky et al., 2009). **(b)** The clean accuracy and 20-step PGD attack (Madry et al., 2017) accuracy on tail classes for the same set of methods. Our method (*yellow star*) surpasses all existing methods, achieving a notable improvement on tail classes.

robustness of head classes, which have a larger number of samples. This highlights the need for more advanced research on adversarial training in long-tailed distributions.

In this paper, we provide an in-depth analysis of why adversarial training in long-tailed distributions is particularly challenging, focusing on the performance on tail classes. Through theoretical analysis, we show that adversarial training causes more severe performance degradation in tail classes compared to natural training. This highlights the inherent difficulty of achieving high robustness in long-tail distributions, especially for the tail classes. Building on these insights, we propose a novel two-step framework designed to improve tail class robustness under adversarial training on long-tailed distribution.

Our framework consists of constructing a balanced dataset from a given unbalanced dataset and employing self-distillation. We first create a sub-dataset where each class contains an equal number of data samples, referred to as the balanced sub-dataset. Then, we adversarially train a self-teacher model on this balanced dataset, achieving higher robustness in tail classes than models trained on the full long-tailed dataset. Subsequently, we apply self-distillation using the balanced self-teacher model to improve tail class performance, resulting in significant gains over baseline models. As shown in Figure 1a, our method achieves the highest accuracy against AutoAttack (Croce & Hein, 2020) and demonstrates significant performance improvements, particularly on tail classes as in Figure 1b. Here are our key contributions:

- We conduct an in-depth analysis to explain why adversarial training on long-tailed datasets results in poor tail class performance. Our findings show that, despite adversarial training, tail class robustness is even lower than natural training.

- Based on these insights, we propose a novel two-step adversarial training approach specifically designed for long-tailed datasets. This method improves upon baselines that merely combine existing long-tailed classification techniques with adversarial training.

- Our approach achieves state-of-the-art performance in adversarial training on long-tailed datasets across various architectures, datasets, and imbalance ratios, leading to significant enhancements in both clean and robust accuracy, with particularly notable improvements on tail classes.

## 2 RELATED WORKS

### 2.1 ADVERSARIAL TRAINING AND DISTILLATION

In response to adversarial attacks (Goodfellow et al., 2014; Carlini & Wagner, 2017; Madry et al., 2017; Athalye et al., 2018), adversarial training (Goodfellow et al., 2014; Madry et al., 2017) empirically stands out as one of the most effective. Adversarial training defines optimization as a min-max problem, where inner maximization generates adversarial inputs, and outer minimization trains the model on these adversarial samples. TRADES (Zhang et al., 2019) incorporates the Kullback-Leibler (KL) divergence loss between the logits of clean and adversarial images. MART (Wang et al., 2020) introduces per-sample weights based on the confidence of each sample. These two methods are used as baseline methods for other recent adversarial training research (Qin et al., 2019; Wu et al., 2020; Bai et al., 2021b; Jin et al., 2022; Tack et al., 2022; Jin et al., 2023; Wei et al., 2023).

The superior performance of adversarial training is primarily observed in large architecture networks, motivating research efforts to improve performance in smaller architectures using techniques such as distillation. Adversarial Robust Distillation (ARD) (Goldblum et al., 2020) proposes a loss function that guides the adversarial output of the student model towards the natural output of the teacher, similar to TRADES (Zhang et al., 2019). Robust Soft Labels Adversarial Distillation (RSLAD) (Zi et al., 2021) leverages teacher logits to improve performance through inner maximization in adversarial training. Many other studies leverage the teacher's logits (Zhu et al., 2021; Maroto et al., 2022; Huang et al., 2023) and gradients (Lee et al., 2023) to distill robustness into the student model.

While these adversarial training and distillation studies have achieved strong robustness, they have only been conducted on balanced datasets where each class has an equal number of samples. This differs significantly from the real-world data configurations we encounter, highlighting the necessity of adversarial training or distillation for unbalanced datasets.

### 2.2 LONG-TAILED RECOGNITION

Extensive research has been conducted to address the performance imbalance inherent in long-tailed distribution datasets. Prominent methods include oversampling the minority tail data (Chawla et al., 2002; Han et al., 2005) and increasing the weight of the minority classes (Cui et al., 2019; Zhang et al., 2021). Although these methods are intuitive, they pose a risk of overfitting on the tail classes and can degrade feature extraction performance (Kang et al., 2019; Zhou et al., 2020). A more effective approach, decoupled learning (Kang et al., 2019; Zhou et al., 2020; Alshammari et al., 2022), separates feature learning from classification to mitigate such issues. Moreover, logit compensation methods have been proposed recently, introducing relatively larger margins between different classes based on prior class frequencies (Cao et al., 2019a; Kang et al., 2019; Menon et al., 2020; Ren et al., 2020; Tan et al., 2020).

### 2.3 LONG-TAILED ADVERSARIAL TRAINING

RoBal (Wu et al., 2021) is the first paper to address adversarial robustness in long-tailed classification. RoBal applies adversarial training with TRADES regularization (Zhang et al., 2019) alongside long-tailed techniques such as balanced softmax (Ren et al., 2020) and class-aware margin (Cao et al., 2019b). Moreover, it provides detailed insights into which modules are effective for long-tailed adversarial training. REAT (Li et al., 2023) aimed to achieve balanced performance by utilizing class-wise weights to generate adversarial examples and expanding the feature space of tail class data. AT-BSL (Yue et al., 2024) revisited the RoBal paper to analyze the necessity of various modules and concluded that only the balanced softmax loss (BSL) is sufficient without the need for complex modules as follows:

$$\mathcal{L}_{BSL}(f(\boldsymbol{x}'), y) = -\log\left(\frac{e^{z_y' + b_y}}{\sum_i e^{z_i' + b_i}}\right), \tag{1}$$

where $\boldsymbol{x}'$ is an adversarially perturbed input of $\boldsymbol{x}$, $z_i' = f(\boldsymbol{x}')_i$ is $i$-th logits of the adversarial input, $b_i = \tau \, log(n_i)$, $\tau$ is a hyperparameter and $n_i$ is the number of examples in the $i$-th class. The

balanced softmax is a commonly used loss in addressing long-tail problems to boost the performance of tail classes (Ren et al., 2020). However, its drawback lies in adjusting the importance of tail classes based on the number of data. In other words, more than balanced softmax is needed to address robustness concerns for tail classes adequately.

There has been no in-depth analysis of robustness degradation in tail classes compared to head classes in adversarially robust long-tailed distributions. In this paper, we aim to improve the overall performance of existing long-tailed adversarial training by achieving sufficient robustness of tail classes.

## 3 ANALYSIS

### 3.1 PRELIMINARY

Let $f$ represent the classification model, which maps the input data space $\mathcal{X}$ to the output label space $\mathcal{Y}$, i.e., $f : \mathcal{X} \rightarrow \mathcal{Y}$. For specific instance of $\mathcal{X}$ and $\mathcal{Y}$, we use $\boldsymbol{x} \in \mathcal{X}$ and $y \in \mathcal{Y}$ and $\boldsymbol{x} = (x_1, x_2, \ldots, x_n)$ where $n$ is the dimension of $\boldsymbol{x}$.

**Definition 1.** *For a classifier $f(\cdot)$, the overall standard error $\mathcal{R}_{std}(\cdot)$ of classifier $f(\cdot)$ is defined as*

$$\mathcal{R}_{std}(f) = \Pr(f(\boldsymbol{x}) \neq y),$$

*and its robust error $\mathcal{R}_{rob}(\cdot)$ is*

$$\mathcal{R}_{rob}(f) = \Pr(\exists \boldsymbol{\delta} \ \text{with} \ \|\boldsymbol{\delta}\|_\infty \leq \epsilon \ s.t. \ f(\boldsymbol{x} + \boldsymbol{\delta}) \neq y)$$

*where $\Pr(\cdot)$ means probability and $\epsilon$ is a non-negative perturbation boundary.*

For simplicity, we denote by $f_{nat}(\cdot)$ the natural classifier that minimizes standard error, and by $f_{rob}(\cdot)$ the robust classifier that minimizes robust error. Additionally, we denote the standard error and robust error for a given class $k$ as $\mathcal{R}_{\text{std}}^k(f)$ and $\mathcal{R}_{\text{rob}}^k(f)$, respectively.

### 3.2 THEORETICAL ANALYSIS

Let the long-tailed dataset for a binary classification task data $\mathcal{S}$ with imbalance ratio $r \geq 1$, *i.e.*, the ratio of the number of instances in the head class ($y = +1$) to the number of instances in the tail class ($y = -1$) is $r$. We assume Gaussian mixture distribution, which is similar to Xu et al. (2021); Lee et al. (2024) as follows.

$$y = \begin{cases} +1, & \text{w.p} \ \frac{r}{r+1} \\ -1, & \text{w.p} \ \frac{1}{r+1} \end{cases} , \quad x_1, \cdots, x_n \stackrel{i.i.d}{\sim} \mathcal{N}(\eta y, 1), \tag{2}$$

where $\eta > \epsilon$ is a constant that determines the standard deviation of the Gaussian distribution. We address a binary classification problem on the above dataset, and then we obtain the following linear function $f_{\boldsymbol{w},b}(\cdot)$, with weight $\boldsymbol{w}$ with bias $b$.

$$f_{\boldsymbol{w},b}(\boldsymbol{x}) = \text{sign}\left(\sum_{k=1}^{n} w_k x_k + b\right). \tag{3}$$

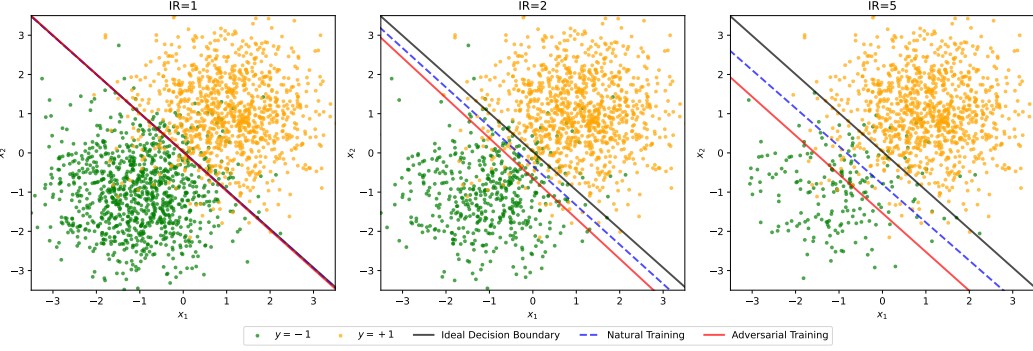

Figure 2: Logistic regression on binary data in Equation (2) with different imbalance ratio (IR).

According to Lemma 1 and Lemma 2 in Appendix A, each weight $w_1, w_2, \cdots, w_n$ of optimal (natural, robust) classifier has the same weight, *i.e.*, $w_1 = w_2 = \cdots = w_n$. We derive the standard and robust error for the tail class of each optimal classifier as follows.

**Theorem 1.** *For a data distribution $\mathcal{S}$, the optimal natural classifier $f_{nat}^*$ and robust classifier $f_{rob}^*$ exhibit the following standard and robust errors for the tail class $-1$ with perturbation margin $0 < \epsilon < \eta$, respectively:*

$$\mathcal{R}_{nat}^{-1}(f_{nat}^*) = \Phi\left(-\sqrt{n}\eta + \frac{\ln r}{2\sqrt{n}\eta}\right), \qquad \mathcal{R}_{rob}^{-1}(f_{nat}^*) = \Phi\left(-\sqrt{n}(\eta - \epsilon) + \frac{\ln r}{2\sqrt{n}\eta}\right), \qquad (4)$$

$$\mathcal{R}_{nat}^{-1}(f_{rob}^*) = \Phi\left(-\sqrt{n}\eta + \frac{\ln r}{2\sqrt{n}(\eta - \epsilon)}\right), \ \mathcal{R}_{rob}^{-1}(f_{rob}^*) = \Phi\left(-\sqrt{n}(\eta - \epsilon) + \frac{\ln r}{2\sqrt{n}(\eta - \epsilon)}\right). \tag{5}$$

A detailed proof of Theorem 1 can be found in Appendix A. From Theorem 1, we can easily infer that both the natural and robust errors of the tail class for both the natural and robust classifiers increase monotonically with respect to the imbalance ratio $r$. Building upon this, we present the following corollary:

**Corollary 1.** *Adversarial training on long-tailed datasets exacerbates the vulnerability of the tail class, making them even less robust than under natural training :*

$$\mathcal{R}_{rob}^{-1}(f_{rob}^*) > \mathcal{R}_{rob}^{-1}(f_{nat}^*). \tag{6}$$

A proof of Corollary 1 is trivial according to eq. (4) and eq. (5) in Theorem 1.

### 3.3 EMPIRICAL ANALYSIS

In Figure 2, we present a visualization of the theoretical analysis in a 2-dimensional space. The data were sampled from Gaussian distributions with $\eta = 1$ and $n = 2$ following Equation (2), considering three different imbalance ratios (IR=1, 2, 5). The figure highlights the decision boundaries formed by both natural and adversarial training ($\epsilon = 0.5$). As the imbalance ratio increases, the decision boundary of the adversarially trained model becomes more distorted, reflecting the model's increased sensitivity to adversarial perturbations in the minority class.

Figure 3 shows the test accuracy of the tail class ($y = -1$) of the logistic regressions. The results indicate a clear trend: as the imbalance ratio grows, the test accuracy for the tail class drops across both natural and adversarial training scenarios. More notably, adversarial training consistently produces more robust errors than natural training. These results align with our theoretical predictions: imbalanced data amplifies clean and robust errors for the tail class, and adversarial training further exacerbates robust errors.

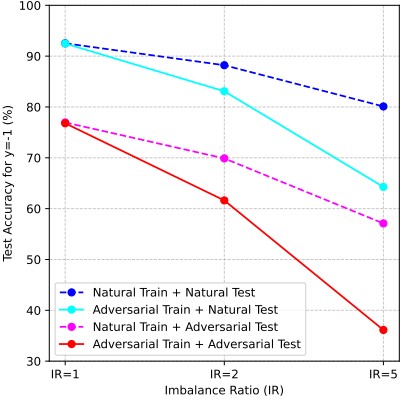

Figure 3: Tail class natural and robust accuracy with respect to natural and adversarial training with different imbalance ratios (IR) in Figure 2.

We further experiment on a long-tailed CIFAR-10 dataset on various imbalance ratios. We conduct both standard and PGD-adversarial training and compare the robust accuracy against PGD attack with $\epsilon = 2/255$ of the entire and tail classes. As shown in Figure 4a, the robust performance of the adversarially-trained model across various imbalance ratios is superior to that of the naturally-trained model, which shows trivial results. However, in Figure 4b, the natural model exhibits better robust performance than the adversarially-trained

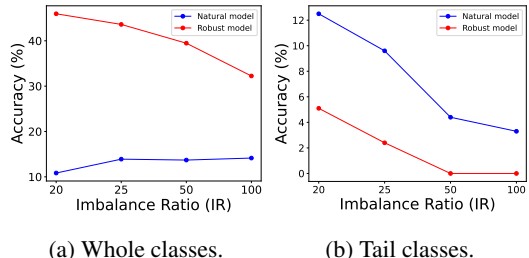

(a) Whole classes.    (b) Tail classes.

Figure 4: Clean and robust accuracy of natural and robust models.

---

**Algorithm 1** Main Algorithm

---

**Input:** Long-tailed dataset $\mathcal{D}$, batch size $N$, epochs $T$, learning rate $\mu$, hyperparameters $\alpha$, $\gamma$,
    and balanced self-teacher training parameters (batch size $N_B$, epochs $T_B$, and learning rate $\mu_B$)
**Output:** Robust model $f$ on long-tailed dataset $\mathcal{D}$

  # balanced self-teacher training
Make balanced dataset $\mathcal{D}_B$ by re-sampling $\mathcal{D}$ with hyperparameter $\gamma$
Randomly initialize $\theta_B$, the weights of balanced self-teacher $f_B$
**for** epoch = 1 to $T_B$ **do**
  **for** mini batch $\{\boldsymbol{x}_i, y_i\}_{i=1}^{N_B}$ in $\mathcal{D}_B$ **do**
    **for** $i = 1$ to $N_B$ **do**
      $\boldsymbol{x}_i' = \text{PGD}(\boldsymbol{x}_i, y_i)$                # PGD attack
    **end for**
    $\theta_B \leftarrow -\mu_B \frac{1}{N_B} \sum_{i=1}^{N_B} \nabla_{\theta_B} \mathcal{L}_{CE}(f_B(\boldsymbol{x}_i'), y_i)$
  **end for**
**end for**

  # main training
Randomly initialize $\theta$, the weights of training model $f$
**for** epoch = 1 to $T$ **do**
  **for** mini batch $\{\boldsymbol{x}_i, y_i\}_{i=1}^{N}$ in $\mathcal{D}$ **do**
    **for** $i = 1$ to $N$ **do**
      $\boldsymbol{x}_i' = \text{PGD}(\boldsymbol{x}_i, y_i)$                # PGD attack
    **end for**
    $\theta \leftarrow -\mu \frac{1}{n} \sum_{i=1}^{n} \nabla_{\theta} \Big[ \mathcal{L}_{BSL}(f(\boldsymbol{x}_i'), y_i)$        # Balanced softmax loss
                        $+\alpha \cdot \mathcal{L}_{KD}(f(\boldsymbol{x}_i'), f_B(\boldsymbol{x}_i)) \Big]$     # Self-distillation
  **end for**
**end for**

---

model in the tail classes. This experiment supports the results of Corollary 1, which clearly demonstrates that while adversarial training generally helps to improve robustness, it could exacerbate performance degradation in the tail classes of long-tailed distributions.

## 4 METHOD

We examine the impact of unbalanced datasets on performance disparity, particularly noting that this disparity becomes more pronounced during robust training compared to natural training. To address this issue, we propose a simple yet effective self-distillation framework.

### 4.1 LIMITATIONS OF EXISTING BALANCED SOFTMAX APPROACHES

Balanced softmax (Wu et al., 2021; Yue et al., 2024) is a powerful method that effectively addresses the issue of tail-class robustness under adversarial training. These works demonstrate that applying Balanced Softmax improves tail-class robustness. However, as shown in Table 6, while Balanced Softmax prevents the robustness of tail classes, it still falls short compared to naive PGD training on a balanced dataset in terms of tail-class robustness. Moreover, our experiments in Table 1, Table 2, and Table 3 demonstrate that tail-class robustness under adversarial attacks remains notably lower than that of head classes. This observation underscores the necessity of additional strategies to explicitly improve tail-class robustness.

### 4.2 TRAINING SELF-TEACHER TO GUIDE TAIL CLASS ROBUSTNESS

To address the limitations of Balanced Softmax and improve tail-class robustness, we construct a balanced sub-dataset $D_B$ by up-sampling tail classes and down-sampling head classes. Specifically, suppose the number of samples of each class in $D$ is $n_1 < n_2 < \cdots < n_C$, then we construct a new dataset where each class contains $\gamma \cdot n_1$ where $\gamma > 1$ is a hyperparameter of adjusting the

Table 1: The clean accuracy and robustness for various algorithms using ResNet-18 on CIFAR-10-LT. T-Clean and T-PGD are clean and PGD-20 accuracy on tail class.

| Method | Best Checkpoint | | | | | Last Checkpoint | | | | |
|--------|------|------|------|---------|-------|------|------|------|---------|-------|
|        | Clean | PGD | AA | T-Clean | T-PGD | Clean | PGD | AA | T-Clean | T-PGD |
| PGD-AT | 52.71 | 29.30 | 27.57 | 12.7 | 1.0 | 56.39 | 26.98 | 25.81 | 20.8 | 2.2 |
| TRADES | 45.79 | 28.66 | 27.01 | 6.7 | 0.8 | 47.10 | 28.00 | 26.45 | 6.4 | 0.6 |
| MART | 44.03 | 29.36 | 27.59 | 5.0 | 0.5 | 47.33 | 28.08 | 26.55 | 10.9 | 1.0 |
| AWP | 51.69 | 32.42 | 30.35 | 5.3 | 0.2 | 51.89 | 32.42 | 30.35 | 10.9 | 0.6 |
| RoBal | 70.54 | 35.33 | 28.83 | 70.4 | 33.1 | **72.80** | 28.04 | 25.00 | 67.7 | 15.9 |
| REAT | 68.34 | 35.98 | 32.45 | 69.5 | 29.1 | 68.32 | 28.67 | 26.68 | 55.7 | 11.6 |
| AT-BSL | 68.43 | 35.87 | 32.27 | 63.2 | 22.0 | 67.60 | 29.40 | 27.46 | 50.1 | 8.7 |
| **Ours** | **70.81** | **38.85** | **34.32** | **73.8** | **36.9** | 71.74 | **37.80** | **33.74** | **74.7** | **36.2** |

number of $D_B$. Using $D_B$, we perform robust training with PGD, resulting in a self-teacher model that is more robust to tail classes compared to models trained on imbalanced datasets. The balanced self-teacher transfers its tail robustness to the student model via adversarial knowledge distillation Lee et al. (2023), as detailed in Algorithm 1. Through this process, the proposed method addresses the insufficient tail-class robustness of Balanced Softmax, enhancing the model's robustness on tail classes while maintaining overall robustness.

# 5 EXPERIMENTS

## 5.1 EXPERIMENT SETTINGS

**Dataset.** We conducted experiments using long-tailed distribution datasets: CIFAR-10-LT, CIFAR-100-LT (Krizhevsky et al., 2009), and Tiny-ImageNet-LT (Le & Yang, 2015), with various imbalance ratios (IR), primarily set at 50 for CIFAR-10-LT, 10 for CIFAR-100-LT and Tiny-ImageNet-LT. Random crop and random horizontal flip were applied, while other augmentations were not utilized unless specified.

**Training details.** We employed ResNet-18 (He et al., 2016a) and WideResNet-34-10 (Zagoruyko & Komodakis, 2016) architectures for CIFAR-10/100-LT, and results for WideResNet-34-10 are included in the appendix. For Tiny-ImageNet-LT, we employed PreActResNet-18 (He et al., 2016b). Initially, we trained a balanced self-teacher using the same model architecture for 30 epochs using a batch size of 32 with a balanced dataset, resampled by the original long-tailed dataset with $\gamma = IR/2$. In the main training phase, we trained for 100 epoch using a batch size of 128 with self-distillation from the balanced self-teacher. We utilized SGD optimization to train both the balanced self-teacher and the main model, setting the learning rate to $0.1$ and weight decay to $5 \times 10^{-4}$. We used an epsilon boundary of $8/255$, a commonly used setting in adversarial training, and employed a 10-step PGD attack during training.

**Comparison models.** As comparison models, we utilized PGD-AT (Madry et al., 2017), TRADES (Zhang et al., 2019), MART (Wang et al., 2020), and AWP (Wu et al., 2020), representing prominent approaches of AT. Additionally, we followed RoBal (Wu et al., 2021), REAT (Li et al., 2023), and AT-BSL (Yue et al., 2024), which focus on long-tailed adversarial training. For long-tailed AT implementation, we meticulously followed the setting of existing methods such as learning rate, batch size, weight decay, etc.

**Evaluation.** Evaluation metrics included clean accuracy, accuracy under a 20-step PGD attack, and AutoAttack (AA) accuracy (Croce & Hein, 2020). Additionally, we assessed clean and 20-step PGD attack accuracy specifically for tail classes, denoted as T-Clean and T-PGD, respectively. In CIFAR-10-LT, the performance evaluation focused on the last class, while CIFAR-100-LT and Tiny-ImageNet-LT evaluated the performance of the tail 10 and 20 classes out of 100 and 200 classes, respectively. We measured performance at both the best and last epoch based on the accuracy under the 20-step PGD attack.

Table 2: The clean accuracy and robustness for various algorithms using ResNet-18 on CIFAR-100-LT. T-Clean and T-PGD are clean and PGD-20 accuracy on the tail class group.

| Method | Best Checkpoint | | | | | Last Checkpoint | | | | |
|---|---|---|---|---|---|---|---|---|---|---|
| | Clean | PGD | AA | T-Clean | T-PGD | Clean | PGD | AA | T-Clean | T-PGD |
| PGD-AT | 42.73 | 17.31 | 15.44 | 20.4 | 7.7 | 43.09 | 15.07 | 14.05 | 22.4 | 6.7 |
| TRADES | 38.83 | 19.05 | 16.06 | 16.5 | 7.3 | 39.63 | 18.86 | 16.18 | 16.1 | 6.8 |
| MART | 38.57 | 19.90 | 17.10 | 16.6 | 7.7 | 40.31 | 17.07 | 15.21 | 19.5 | 7.2 |
| AWP | 40.46 | 21.85 | 18.58 | 16.2 | 8.5 | 40.15 | 21.71 | 18.33 | 16.2 | 8.6 |
| RoBal | 44.27 | 19.67 | 16.78 | 18.4 | 8.0 | 46.46 | 16.28 | 14.73 | 23.3 | 6.7 |
| REAT | 45.73 | 18.22 | 15.82 | 32.2 | 11.4 | 45.53 | 15.64 | 14.27 | 33.0 | 10.5 |
| AT-BSL | 45.38 | 18.04 | 15.73 | 33.1 | 12.4 | 45.48 | 15.36 | 14.07 | 31.5 | 9.1 |
| **Ours** | **46.13** | **22.42** | **18.73** | **38.9** | **17.9** | **47.22** | **21.82** | **18.53** | **37.9** | **17.6** |

Table 3: The clean accuracy and robustness for various algorithms using PreActResNet-18 on Tiny-ImageNet-LT.

| Method | Best Checkpoint | | | | | Last Checkpoint | | | | |
|---|---|---|---|---|---|---|---|---|---|---|
| | Clean | PGD | AA | T-Clean | T-PGD | Clean | PGD | AA | T-Clean | T-PGD |
| PGD-AT | 34.89 | 14.17 | 10.98 | 15.8 | 5.4 | 36.55 | 9.69 | 8.45 | 23.2 | 5.0 |
| TRADES | 33.76 | 13.71 | 10.00 | 15.4 | 6.4 | 32.97 | 12.50 | 9.61 | 16.0 | 6.0 |
| MART | 31.15 | 15.45 | 11.94 | 14.4 | 6.0 | 32.91 | 12.42 | 10.32 | 17.8 | 7.6 |
| AWP | 32.28 | 15.09 | 11.27 | 14.2 | 5.6 | 32.13 | 13.95 | 11.10 | 14.4 | 6.6 |
| RoBal | 35.25 | 14.01 | 10.44 | 15.0 | 4.4 | 37.97 | 10.51 | 8.64 | 21.8 | 4.8 |
| REAT | 38.37 | 15.25 | 11.99 | 33.0 | 12.6 | 38.48 | 10.58 | 9.07 | 31.6 | 8.8 |
| AT-BSL | 38.38 | 15.39 | 11.85 | 30.8 | 13.0 | 38.41 | 10.25 | 8.90 | 32.4 | 7.0 |
| **Ours** | **38.44** | **17.02** | **12.57** | **36.6** | **16.0** | **49.37** | **14.09** | **11.15** | **33.8** | **12.6** |

## 5.2 MAIN RESULTS

We demonstrated excellent performance across all datasets, including CIFAR-10-LT, CIFAR-100-LT, and Tiny-ImageNet-LT in Table 1, Table 2, and Table 3, respectively. The experimental results on the WideResNet-34-10 architecture can be found in Table 7and Table 8 in the appendix. Particularly noteworthy is the substantial improvement in performance for tail classes. When adversarial training methods such as PGD-AT, TRADES, MART, and AWP are naively applied to long-tailed datasets, overall performance remains reasonable compared to Robal, REAT, AT-BSL, but performance for the tail classes notably suffers. For instance, while AWP exhibits superior performance compared to RoBal, the clean accuracy and robust accuracy for tail classes are significantly low. Long-tailed adversarial training methods such as RoBal, REAT, and AT-BSL show considerable improvement in tail class performance compared to other adversarial training methods. However, when compared to the performance of the entire class, it is still evident that the performance remains imbalanced. In contrast, our method shows significant improvement in the performance of the tail classes, resulting in minimal difference compared to the performance of the entire classes. Additionally, we achieved overall better performance than the baseline at both the best and last checkpoints.

## 5.3 ABLATION

In this section, we conduct further experiments to corroborate our main contribution.

### 5.3.1 AUGMENTATION

Following the inclusion of various augmentation experiments outlined in AT-BSL, we conducted experiments applying RandAugment (RA) (Cubuk et al., 2020) and AutoAugment (AuA) (Cubuk et al., 2019) in Table 4. While applying augmentation led to overall performance improvements, the

Table 4: The clean accuracy and robustness with augmentation using ResNet-18 on CIFAR-100-LT. T-Clean and T-PGD are clean and 20-step PGD accuracy on the tail class group.

| Method | Best Checkpoint | | | | | Last Checkpoint | | | | |
|---|---|---|---|---|---|---|---|---|---|---|
| | Clean | PGD | AA | T-Clean | T-PGD | Clean | PGD | AA | T-Clean | T-PGD |
| Robal | 44.27 | 19.67 | 16.78 | 18.4 | 8.0 | 46.46 | 16.28 | 14.73 | 23.3 | 6.7 |
| Robal-RA | 44.64 | 20.11 | 17.02 | 15.8 | 7.7 | 47.62 | 18.63 | 16.05 | 19.2 | 7.4 |
| Robal-AuA | 45.87 | 20.24 | 17.05 | 17.4 | 6.7 | 47.42 | 19.32 | 16.30 | 18.2 | 7.4 |
| Reat | 45.38 | 18.04 | 15.73 | 33.1 | 12.4 | 45.48 | 15.36 | 14.07 | 31.5 | 9.1 |
| Reat-RA | 46.94 | 21.71 | 18.02 | 33.3 | 14.9 | 50.41 | 20.33 | 17.46 | 36.6 | 14.7 |
| Reat-AuA | 47.86 | 23.09 | 19.43 | 34.0 | 16.7 | 50.56 | 22.20 | 18.60 | 36.5 | 16.5 |
| AT-BSL | 45.38 | 18.04 | 15.73 | 33.1 | 12.4 | 45.48 | 15.36 | 14.07 | 31.5 | 9.1 |
| AT-BSL-RA | 48.38 | 22.18 | 18.58 | 34.7 | 16.8 | 50.33 | 20.29 | 17.42 | 37.1 | 14.6 |
| AT-BSL-AuA | 47.30 | 22.78 | 18.66 | 34.2 | 16.4 | 50.57 | 21.98 | 18.45 | 36.8 | 16.3 |
| **Ours** | 46.13 | 22.42 | 18.73 | **38.9** | 17.9 | 47.22 | 21.82 | 18.53 | 37.9 | 17.6 |
| **Ours-RA** | 48.78 | 23.58 | 19.30 | 34.9 | 17.2 | **50.98** | 22.43 | 18.80 | **38.2** | 16.9 |
| **Ours-AuA** | **50.14** | **24.60** | **20.08** | 36.4 | **18.0** | 50.46 | **24.32** | **20.28** | 37.6 | **18.5** |

Table 5: The clean accuracy and robustness with different Imbalance Ratio(IR) using ResNet-18 on CIFAR-100-LT. T-Clean and T-PGD are clean and 20-step of PGD accuracy on the tail class group.

| IR | Method | Best Checkpoint | | | | | Last Checkpoint | | | | |
|---|---|---|---|---|---|---|---|---|---|---|---|
| | | Clean | PGD | AA | T-Clean | T-PGD | Clean | PGD | AA | T-Clean | T-PGD |
| 50 | RoBal | 33.52 | 14.56 | 12.27 | 1.9 | 0.9 | 34.81 | 12.16 | 11.02 | 4.6 | 1.7 |
| | REAT | 26.62 | 13.73 | 10.64 | 9.2 | 4.1 | 36.51 | 12.28 | 11.15 | 19.1 | 4.4 |
| | AT-BSL | 30.06 | 13.80 | 10.91 | 10.3 | 4.2 | 36.46 | 12.07 | 11.21 | 17.8 | 4.4 |
| | **Ours** | **38.09** | **16.65** | **13.58** | **14.8** | **5.2** | **38.56** | **16.08** | **13.52** | **19.2** | **5.8** |
| 20 | RoBal | 40.08 | 16.91 | 14.28 | 12.1 | 5.4 | 41.28 | 13.96 | 12.70 | 15.5 | 5.1 |
| | REAT | 33.17 | 15.82 | 13.01 | 20.0 | 9.2 | 41.73 | 13.79 | 12.58 | 29.8 | 7.7 |
| | AT-BSL | 41.70 | 15.51 | 13.62 | 30.7 | 10.4 | 41.41 | 13.49 | 12.48 | 27.1 | 7.9 |
| | **Ours** | **42.54** | **19.66** | **16.36** | **33.2** | **13.3** | **42.64** | **19.24** | **15.97** | **32.5** | **13.7** |
| 10 | RoBal | 44.27 | 19.67 | 16.26 | 16.7 | 7.6 | 46.46 | 16.28 | 14.73 | 23.3 | 6.7 |
| | REAT | 45.73 | 18.22 | 15.82 | 34.4 | 12.2 | 45.53 | 15.64 | 14.27 | 33.0 | 10.5 |
| | AT-BSL | 45.38 | 18.04 | 15.73 | 33.1 | 12.4 | 45.48 | 15.36 | 14.07 | 31.5 | 9.1 |
| | **Ours** | **47.22** | **21.82** | **18.53** | **37.9** | **17.6** | **47.22** | **21.82** | **18.53** | **37.9** | **17.6** |
| 5 | RoBal | 49.49 | 21.66 | 18.59 | 26.1 | 11.9 | 51.56 | 18.15 | 16.58 | 34.7 | 9.9 |
| | REAT | 49.48 | 21.95 | 18.98 | 39.3 | 18.3 | 49.76 | 18.19 | 16.65 | 40.7 | 14.1 |
| | AT-BSL | 49.75 | 21.53 | 18.65 | 42.3 | 18.2 | 49.41 | 18.12 | 16.56 | 40.0 | 13.7 |
| | **Ours** | **50.77** | **24.13** | **20.10** | **44.0** | **20.9** | **51.92** | **25.00** | **21.14** | **43.8** | **20.4** |

best results were achieved when augmentation was applied to our method. Our method consistently outperformed baselines on robustness with augmentation setting including tail class performance with augmentation.

### 5.3.2 DIFFERENT IMBALANCE RATIO

In Table 5, we conducted experiments using different imbalance ratios (IR). As the IR increases, the number of tail classes decreases, leading to decreased robustness. In all cases, our method outperforms the baseline in terms of both overall and tail robustness. This indicates that our proposed framework generally performs well across different IR settings.

Table 6: The clean accuracy and robustness using ResNet-18 on CIFAR-100-LT. T-Clean and T-PGD represent clean and 20-step PGD accuracy on a tail class.

| Dataset | Method | Clean | PGD | T-Clean | T-PGD |
|---|---|---|---|---|---|
| $D$ | Robal | 44.27 | **19.67** | 18.4 | 8.0 |
|  | Reat | **45.73** | 18.22 | 32.2 | 11.4 |
|  | BSL | 45.38 | 18.04 | 33.1 | 12.4 |
| $D_B$ | PGD-AT | 35.71 | 14.94 | **34.6** | **14.1** |

### 5.3.3 EFFECT OF BALANCED SUBSET

To evaluate the effect of the balanced subset $D_B$, we compare the performance of models trained with PGD-AT on $D_B$ against Robal, Reat, and BSL trained on $D$, which incorporate techniques like balanced softmax to address long-tailed distributions. For simplicity, we denote PGD-AT$_{D_B}$ as the model trained with PGD-AT on the balanced subset $D_B$.

As shown in Table 6, while the PGD-AT model trained on $D_B$ achieved the lowest overall performance, it demonstrated the best results for tail classes, T-Clean and T-PGD. This suggests that even with fewer training epochs, the balanced subset is effective for improving performance on tail classes. In Figure 5, we apply the same methods with different teachers where the performance is summarized in Table 6. Interestingly, the best results were achieved when we utilize PGD-AT$_{D_B}$ as the teacher model, despite having the lowest overall performance. Specifically, the performance on the tail classes highlights the effectiveness of the teacher trained on the balanced subset, as it demonstrates superior performance on the tail class compared to other methods. This underscores the utility of the balanced subset in improving tail class performance. Additionally, although we trained the teacher for self-distillation using a simple method, PGD-AT, developing a more effective teacher remains an area for future work.

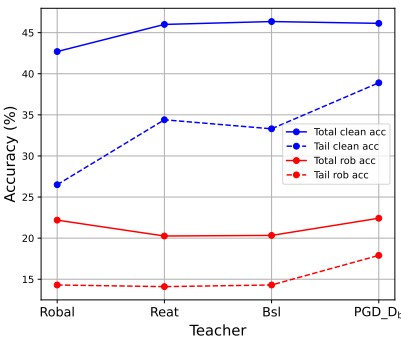

Figure 5: Clean and robust accuracy according to different teachers. Robal, Reat, and Bsl were trained with 100 epochs, while PGD-AT$_{D_B}$ used a teacher trained with 30 epochs.

## 6 CONCLUSION

Building on the observation that adversarial training methods inherently struggle with tail classes, we propose effective strategies to address the lower performance on these classes. We first train a balanced self-teacher and subsequently perform knowledge distillation from this self-teacher. This approach leads to significant improvements in long-tailed adversarial training, enhancing both overall robustness and tail class robustness.

**Discussion** It is well known that adversarial training varies in difficulty across classes, and performance also differs by class. This presents a fairness issue, indicating that in robustness, not only the number of data points but also the intrinsic difficulty of each class plays a role. While this paper focuses solely on data quantity, designing robust models that account for class-level fairness remains an area for future work.

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

## A  THEORITICAL PROOF

**Lemma 1.** *Given the data distribution $\mathcal{S}$, an optimal natural classifier that minimizes the overall standard error has optimal weight that satisfies $w_1 = w_2 = \cdots = w_n$.*

*Proof.* Let's assume, for the sake of contradiction, that the optimal weights do not satisfy the given condition. In other words, for some $i \neq j$ and $i, j \in \{1, 2, \cdots, n\}$, we assume if there exist $w_i < w_j$. Then, we obtain the following standard error

$$
\begin{aligned}
\mathcal{R}_{nat}(f) = \Pr(y = -1) \cdot \Pr\left( \sum_{k \neq i, k \neq j}^{n} w_k \mathcal{N}(-\eta, 1) + b + w_i \mathcal{N}(-\eta, 1) + w_j \mathcal{N}(-\eta, 1) > 0 \right) \\
+ \Pr(y = +1) \cdot \Pr\left( \sum_{k \neq i, k \neq j}^{n} w_k \mathcal{N}(+\eta, 1) + b + w_i \mathcal{N}(+\eta, 1) + w_j \mathcal{N}(+\eta, 1) < 0 \right)
\end{aligned}
\tag{7}
$$

However, if we define a new classifier $f'$, which has the same weight vector as classfier $f$ but uses $w_j$ to replace $w_i$. The resulting standard error for the new classifier $f'$ can be obtained as follows

$$
\begin{aligned}
\mathcal{R}_{nat}(f') = \Pr(y = -1) \cdot \Pr\left( \sum_{k \neq i, k \neq j}^{n} w_k \mathcal{N}(-\eta, 1) + b + w_j \mathcal{N}(-\eta, 1) + w_j \mathcal{N}(-\eta, 1) > 0 \right) \\
+ \Pr(y = +1) \cdot \Pr\left( \sum_{k \neq i, k \neq j}^{n} w_k \mathcal{N}(+\eta, 1) + b + w_j \mathcal{N}(+\eta, 1) + w_j \mathcal{N}(+\eta, 1) < 0 \right)
\end{aligned}
\tag{8}
$$

Given that $w_i < w_j$, the $f'$ has a smaller error than $f$, which contradicts the assumption that $f$ is the optimal classifier with the least error. $\qquad\square$

**Lemma 2.** *Given the data distribution $\mathcal{S}$, an optimal robust classifier that minimizes the robust error has optimal weight that satisfies $w_1 = w_2 = \cdots = w_n$.*

Similar to the Lemma 1, it can be easily proved with the same argument.

### A.1  PROOF OF THEOREM 1

*Proof.* By the Lemma 1, the optimal classifier $f_{nat}$ for standard error has optimal weight of $w_1 = w_2 = \cdots = w_n$. For simplicity, we assume l2-norm of $\boldsymbol{w} = 1$, *i.e.*, $\boldsymbol{w} = (1/\sqrt{n}, 1/\sqrt{n}, \ldots, 1/\sqrt{n})$. following existing works Xu et al. (2021); Lee et al. (2024). Then, the standard errors of $f_{nat}$ can

be formulated as follows.

$$\mathcal{R}_{nat}(f_{nat}) = \Pr(y = +1) \cdot \mathcal{R}_{nat}^{+1}(f_{nat}) + \Pr(y = -1) \cdot \mathcal{R}_{nat}^{-1}(f_{nat})$$

$$= \frac{r}{r+1} \Pr\left(f(\boldsymbol{x}) \neq y | y = +1\right) + \frac{1}{r+1} \Pr\left(f(\boldsymbol{x}) \neq y | y = -1\right)$$

$$= \frac{r}{r+1} \Pr\left(\sum_{k=1}^{n} \frac{1}{\sqrt{n}} \mathcal{N}(+\eta, 1) + b < 0\right) + \frac{1}{r+1} \Pr\left(\sum_{k=1}^{n} \frac{1}{\sqrt{n}} \mathcal{N}(-\eta, 1) + b > 0\right)$$

$$= \frac{r}{r+1} \cdot \Phi(-\sqrt{n}\eta - b) + \frac{1}{r+1} \cdot \Phi(-\sqrt{n}\eta + b) \tag{9}$$

Here, $\Phi$ represents the cumulative distribution function of the standard normal distribution. To determine the optimal value of $b$, we solve the equation $d\mathcal{R}_{nat}(f_{nat})/db = 0$.

$$\frac{d\mathcal{R}_{nat}(f_{nat})}{db} = -\frac{r}{r+1} \cdot \phi(-\sqrt{n}\eta - b) + \frac{1}{r+1} \cdot \phi(-\sqrt{n}\eta + b) = 0$$

$$-r \cdot \phi(-\sqrt{n}\eta - b) + \phi(-\sqrt{n}\eta + b) = 0$$

$$-r \cdot \exp\left(-\frac{1}{2}(-\sqrt{n}\eta - b)^2\right) + \exp\left(-\frac{1}{2}(-\sqrt{n}\eta + b)^2\right) = 0 \tag{10}$$

Here, $\phi$ represents the standard normal distribution function. Therefore, the optimal $b_{nat}^*$ for natural classifier is follows,

$$b_{nat}^* = \frac{\ln r}{2\sqrt{n}\eta}. \tag{11}$$

By using the optimal natural classifier, the standard error of the tail class can be formulated as follows,

$$\mathcal{R}_{nat}^{-1}(f_{nat}^*) = \Phi\left(-\sqrt{n}\eta + \frac{\ln r}{2\sqrt{n}\eta}\right). \tag{12}$$

Then, the robust error of the tail class with optimal natural classifier can be calculated as follows,

$$\mathcal{R}_{rob}^{-1}(f_{nat}^*) = \Pr(\exists \boldsymbol{\delta} \text{ with } \|\boldsymbol{\delta}\|_{\infty} \leq \epsilon \text{ s.t. } f_{nat}^*(\boldsymbol{x} + \boldsymbol{\delta}) \neq y | y = -1)$$

$$= \Pr\left(\sum_{k=1}^{n} \frac{1}{\sqrt{n}}(x_i + \epsilon) + b_{nat}^* > 0\right)$$

$$= \Pr\left(\sum_{k=1}^{n} \frac{1}{\sqrt{n}} \mathcal{N}(-\eta + \epsilon, 1) + b_{nat}^* > 0\right)$$

$$= \Pr\left(\mathcal{N}(0, 1) < -\sqrt{n}(\eta - \epsilon) + b_{nat}^*\right)$$

$$= \Phi\left(-\sqrt{n}(\eta - \epsilon) + \frac{\ln r}{2\sqrt{n}\eta}\right). \tag{13}$$

Similarly, based on the Lemma 2, the optimal classifier $f_{rob}$ for robust error has optimal weight of $w_1 = w_2 = \cdots = w_n = 1/\sqrt{n}$. Therefore, the robust errors of $f_{rob}$ can be formulated as follows with adversarial noise $\epsilon$ satisfying $0 < \epsilon < \eta$

$$\mathcal{R}_{rob}(f_{rob}) = \Pr(y = +1) \cdot \mathcal{R}_{rob}^{+1}(f_{rob}) + \Pr(y = -1) \cdot \mathcal{R}_{rob}^{-1}(f_{rob})$$

$$= \frac{r}{r+1} \cdot \Pr\left(\sum_{k=1}^{n} \frac{1}{\sqrt{n}} \mathcal{N}(+\eta - \epsilon, 1) + b < 0\right)$$

$$+ \frac{1}{r+1} \cdot \Pr\left(\sum_{k=1}^{n} \frac{1}{\sqrt{n}} \mathcal{N}(-\eta + \epsilon, 1) + b > 0\right)$$

$$= \frac{r}{r+1} \cdot \Pr\left(\mathcal{N}(0, 1) < -\sqrt{n}(\eta - \epsilon) - b\right)$$

$$+ \frac{1}{r+1} \cdot \Pr\left(\mathcal{N}(0, 1) < -\sqrt{n}(\eta - \epsilon) + b\right)$$

$$= \frac{r}{r+1} \cdot \Phi(-\sqrt{n}(\eta - \epsilon) - b) + \frac{1}{r+1} \cdot \Phi(-\sqrt{n}(\eta - \epsilon) + b) \tag{14}$$

To determine the optimal value of $b$, we solve the equation $d\mathcal{R}_{rob}(f_{rob})/db = 0$.

$$\frac{d\mathcal{R}_{rob}(f_{rob})}{db} = -\frac{r}{r+1} \cdot \phi(-\sqrt{n}(\eta - \epsilon) - b) + \frac{1}{r+1} \cdot \phi(-\sqrt{n}(\eta - \epsilon) + b) = 0$$

$$-r \cdot \phi(-\sqrt{n}(\eta - \epsilon) - b) + \phi(-\sqrt{n}(\eta - \epsilon) + b) = 0$$

$$-r \cdot \exp\left(-\frac{1}{2}(-\sqrt{n}(\eta - \epsilon) - b)^2\right) + \exp\left(-\frac{1}{2}(-\sqrt{n}(\eta - \epsilon) + b)^2\right) = 0 \quad (15)$$

Therefore, the optimal $b_{rob}^*$ for robust classifier is follows,

$$b_{rob}^* = \frac{\ln r}{2\sqrt{n}(\eta - \epsilon)}. \quad (16)$$

Then, the standard and robust error of the tail class with optimal robust classfier can be formulated as follows,

$$\mathcal{R}_{nat}^{-1}(f_{rob}^*) = \Phi\left(-\sqrt{n}\eta + \frac{\ln r}{2\sqrt{n}(\eta - \epsilon)}\right), \quad (17)$$

$$\mathcal{R}_{rob}^{-1}(f_{rob}^*) = \Phi\left(-\sqrt{n}(\eta - \epsilon) + \frac{\ln r}{2\sqrt{n}(\eta - \epsilon)}\right). \quad (18)$$

$\square$

## B ADDITIONAL EXPERIMENTS

### B.1 EXPERIMENTS ON ANOTHER ARCHITECTURE.

We conducted experiments using WideResNet-34-10. Similar to the results of ResNet-18 in the main paper, our method consistently demonstrated superior performance. Notably, on both CIFAR-10-LT and CIFAR-100-LT datasets, significant performance improvements were observed in both T-Clean and T-PGD settings. While RoBal exhibited a marginally higher clean accuracy in the case of the best checkpoint on CIFAR-10-LT, the difference compared to our method is negligible. However, our method achieved approximately a 5% point improvement in robust accuracy against auto attack on CIFAR-10-LT. In the CIFAR-100-LT dataset, our method demonstrated the best performance in terms of both clean accuracy and robustness across all classes. Additionally, the improvements in T-Clean and T-PGD demonstrate that our method is more suitable for handling long-tail distributions.

Table 7: The clean accuracy and robustness for various algorithms using WideResNet-34-10 on CIFAR-10-LT.

| Method | Best Checkpoint | | | | | Last Checkpoint | | | | |
|---|---|---|---|---|---|---|---|---|---|---|
| | Clean | PGD | AA | T-Clean | T-PGD | Clean | PGD | AA | T-Clean | T-PGD |
| PGD-AT | 58.86 | 30.57 | 29.43 | 18.5 | 2.1 | 59.10 | 26.3 | 25.66 | 19.0 | 1.9 |
| TRADES | 51.93 | 30.45 | 29.20 | 4.5 | 0.3 | 55.36 | 27.62 | 26.99 | 19.2 | 2.9 |
| MART | 48.92 | 31.45 | 29.85 | 9.5 | 0.9 | 54.81 | 27.25 | 26.29 | 23.1 | 2.0 |
| AWP | 51.69 | 32.42 | 30.35 | 5.3 | 0.2 | 51.89 | 29.19 | 27.45 | 10.9 | 0.6 |
| RoBal | **74.46** | 32.82 | 25.72 | 71.5 | 22.8 | 70.03 | 24.74 | 23.09 | 50.6 | 5.7 |
| REAT | 73.16 | 33.45 | 28.71 | 66.4 | 20.8 | 64.11 | 25.90 | 25.00 | 31.7 | 3.6 |
| AT-BSL | 73.23 | 35.08 | 32.26 | 66.4 | 18.9 | 66.23 | 26.87 | 25.98 | 40.6 | 4.3 |
| **Ours** | 73.97 | **39.25** | **35.97** | **74.3** | **33.7** | **72.38** | **31.15** | **29.10** | **60.4** | **12.7** |

### B.2 ADDITIONAL EXPERIMENT OF MORE TRAINING EPOCHS

Since we employed additional training epochs for self-distillation, we also trained the baselines with more epochs and summarized the results in Table 9. The results showed that increasing the training epochs for the baselines did not lead to performance improvements; in REAT, performance even declined when more training epochs were utilized. As a result, it is clear that the efficacy of our approach is not solely a consequence of increasing the number of training epochs.

Table 8: The clean accuracy and robustness for various algorithms using WideResNet-34-10 on CIFAR-100-LT.

| Method | Best Checkpoint | | | | | Last Checkpoint | | | | |
|---|---|---|---|---|---|---|---|---|---|---|
| | Clean | PGD | AA | T-Clean | T-PGD | Clean | PGD | AA | T-Clean | T-PGD |
| PGD-AT | 47.48 | 19.36 | 17.79 | 25.4 | 8.7 | 46.09 | 16.51 | 15.67 | 24.3 | 7.2 |
| TRADES | 42.67 | 20.89 | 18.42 | 18.3 | 6.9 | 43.99 | 18.53 | 17.51 | 19.9 | 7.4 |
| MART | 41.54 | 21.52 | 18.83 | 19.2 | 9.2 | 43.08 | 17.00 | 15.84 | 22.8 | 7.8 |
| AWP | 45.53 | 23.23 | 19.92 | 20.4 | 7.9 | 47.05 | 21.97 | 19.21 | 23.1 | 8.8 |
| RoBal | 49.06 | 18.23 | 16.79 | 27.6 | 9.4 | 46.92 | 15.48 | 14.69 | 28.0 | 6.8 |
| REAT | 49.06 | 20.00 | 18.08 | 34.4 | 12.2 | 47.65 | 16.95 | 15.60 | 33.6 | 9.8 |
| AT-BSL | 50.05 | 18.96 | 17.10 | 38.3 | 13.3 | 47.95 | 16.40 | 15.31 | 32.2 | 9.5 |
| **Ours** | **50.55** | **23.43** | **20.16** | **38.4** | **19.5** | **50.87** | **22.21** | **19.44** | **42.5** | **18.4** |

Table 9: The clean accuracy and robustness for various algorithms using ResNet-18 on CIFAR-100-LT. T-Clean and T-PGD are clean and PGD-20 accuracy on the tail class group.

| Method | Best Checkpoint | | | | | Last Checkpoint | | | | |
|---|---|---|---|---|---|---|---|---|---|---|
| | Clean | PGD | AA | T-Clean | T-PGD | Clean | PGD | AA | T-Clean | T-PGD |
| RoBal (100 epochs) | 44.27 | 19.67 | 16.78 | 18.4 | 8.0 | 46.46 | 16.28 | 14.73 | 23.3 | 6.7 |
| RoBal (200 epochs) | 44.20 | 19.70 | 17.01 | 17.5 | 8.1 | 45.60 | 15.06 | 14.03 | 24.1 | 6.7 |
| REAT (100 epochs) | 45.73 | 18.22 | 15.82 | 32.2 | 11.4 | 45.53 | 15.64 | 14.27 | 33.0 | 10.5 |
| REAT (200 epochs) | 44.67 | 16.48 | 14.53 | 29.6 | 10.3 | 44.80 | 14.86 | 13.64 | 28.9 | 7.9 |
| AT-BSL (100 epochs) | 45.38 | 18.04 | 15.73 | 33.1 | 12.4 | 45.48 | 15.36 | 14.07 | 31.5 | 9.1 |
| AT-BSL (200 epochs) | 45.01 | 17.19 | 14.56 | 28.9 | 9.7 | 44.04 | 14.23 | 13.23 | 28.5 | 7.2 |
| **Ours** | **46.13** | **22.42** | **18.73** | **38.9** | **17.9** | **47.22** | **21.82** | **18.53** | **37.9** | **17.6** |

(a) Clean accuracy

(b) Robust accuracy

Figure 6: Hyperparmeter sensitivity of entire class

(a) Clean accuracy

(b) Robust accuracy

Figure 7: Hyperparmeter sensitivity of tail class

## B.3 SENSITIVITY OF HYPERPARAMETER

In Figure 6, we experiment with the sensitivity of the self-distillation weight parameter, $\alpha$, and the sampling rate, $\gamma$, where $r$ is an imbalance ratio between the class with the largest number of samples and the class with the smallest number of samples. We can see that as $\alpha$ increases, robustness improves, but clean accuracy slightly decreases. This indicates a trade-off between robustness and clean accuracy, which is expected given the use of adversarial distillation techniques. In the case of $\gamma$, it did not significantly impact performance. However, when $\gamma$ is larger—meaning more samples are used to train the self-teacher—both clean accuracy and robustness showed improvement.

In Figure 7, we plot the tail class performance. In this case, we observed that as $\alpha$ increases, *i.e.*, as the weight of the loss for knowledge distillation from the balanced self-teacher increases, the clean and robust performance of the tail class improves. The sensitivity to $\gamma$ was not significant.

## B.4 VARIANCE OF MULTIPLE RUNS

Table 10: The clean accuracy and robustness for various algorithms using ResNet18 on CIFAR-100-LT.

| Runs | Clean | PGD | T-Clean | T-PGD |
|---|---|---|---|---|
| 1 | 46.13 | 22.42 | 38.9 | 17.9 |
| 2 | 46.57 | 22.23 | 37.5 | 17.7 |
| 3 | 46.47 | 22.27 | 37.8 | 17.8 |
| 4 | 46.59 | 22.12 | 38.8 | 17.8 |
| 5 | 46.01 | 22.52 | 38.9 | 18.0 |
| Average | 46.35 | 22.31 | 38.38 | 17.84 |
| Standard deviation | 0.27 | 0.15 | 0.68 | 0.11 |

In Table 10, we conducted five experiments and computed the mean and standard deviation to assess the impact of randomness. The results show that the standard deviations are relatively small, indicating that the model's performance is consistent across different runs. This suggests that the observed improvements are robust and not significantly influenced by random fluctuations in the training process.

## B.5 CLASS-WISE ROBUSTNESS.

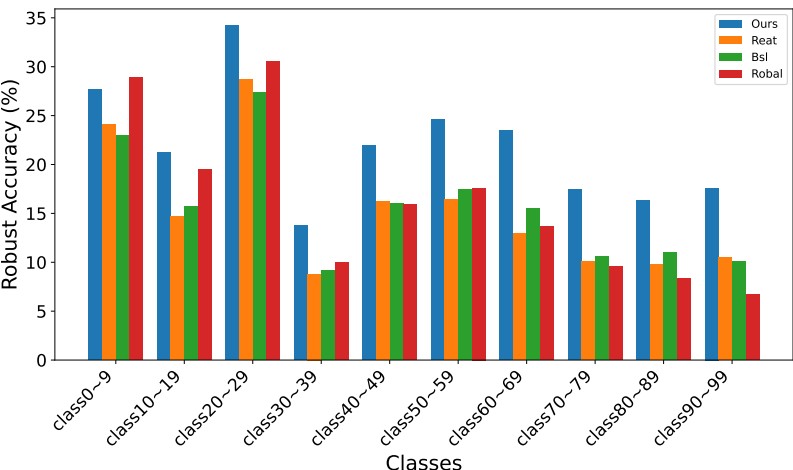

Figure 8: Class-wise robustness.

In Figure 8, we divided the classes of CIFAR-100 into 10 groups and measured the robustness across them. As we move from class 0 to class 99, the number of data points decreases. Our

method demonstrated higher robustness across all class groups compared to the baseline. Notably, it achieved the best performance in all groups except the first group. In contrast, Robal showed strong performance on the first group (head classes) but the worst performance on the last group (tail classes).

