# OpenReview forum: "Long-tailed Adversarial Training with Self-Distillation"
_ICLR.cc/2025/Conference — ICLR 2025 Poster_

### Official Review · Reviewer_Uh2k · 2024-11-01

**Soundness:** 3
**Presentation:** 2
**Contribution:** 3
**Rating:** 6
**Confidence:** 4

**Summary:**

The paper's main contribution is the two-phase adversarial training of deep neural networks (ResNet-18) on CIFAR 10/100 and TynyImageNet datasets. Phase one is training on a Balanced dataset (e.g., 30 epochs), and Phase two is the Knowledge distillation on the long-tailed imbalanced dataset to improve the robustness of the model on a long-tailed imbalanced dataset.

For this contribution, the paper presents a theoretical analysis of Theorem 1, which says robustness training will further impact (worsen) the accuracy of models on long-tailed datasets. Most of the iterative analysis is in the part of the Appendix rather than the main part of the paper. The proof of Corollary indicates that it is trivial according to equations 4 and 5, which appears to indicate that the readers should figure it out themselves.

The main results show improvement compared to some SOTA algorithms. The author's results are shown to be the best in all cases. A typical training set was used for these results.

**Strengths:**

The two-phase training methodology uses the "self-distillation" approach, where phase one is trained on balanced datasets, and this trained model is used for knowledge distillation in the second phase to make models robust on long-trailed datasets. The performance of this approach is shown to be better performance. That is, the results shown surpass the mentioned SOTA results on RestNet18.

**Weaknesses:**

Explain why the proof of the theorem is not generic for any iterative learning processes. Also, a step-by-step explanation of how the weight values affect robustness accuracy in the proof is required.

Include a visual or mathematical explanation connecting Figure 2 and Theorem 1 to clarify their relationship.

The definition of variables $\Phi$ and $r$ in Equations 4 and 5 are required.

For reproducibility, the authors should indicate the availability of codes in the experimentation section.

**Questions:**

Mention which dataset is used for  Figure 3.

A comparison with balanced datasets to determine where adversarial training improves the performance of adversarial tests could be included in Figure 3.

In Figure 3, at IR =1, explain why both natural and adversarial training produces the same test results on the adversarial test data.

Explain what loss function is used for $L_{KD}$ in Algorithms 1were used.

---

> ### Author Response · Authors · 2024-11-19
> **Reply to Reviewer Uh2k. (1/2)**
>
> Thank you for your detailed feedback and constructive comments. We appreciate your recognition of our two-phase training methodology and its performance improvements over existing SOTA results.
>
> **W1-1. Proof of Theorem and Generalization to Iterative Learning**
>
> **A.** Providing a fully theoretical proof that generalizes to all iterative learning processes is indeed highly challenging. While such a general proof would be ideal, we focused on a more foundational setting with a simplified linear model to illustrate the impact of adversarial training on tail-class performance. Our theorem demonstrates that even in this basic setup, adversarial training disproportionately affects tail classes, suggesting that similar vulnerabilities may also be present in larger, more complex models.
> To supplement this theoretical approach, we validated our findings experimentally. As shown in Section 3.3, Figure 4 demonstrates similar results on a ResNet-18 model trained with CIFAR-10-LT, aligning with our theorem’s predictions. This experimental evidence reinforces the applicability of our theoretical insights to more complex architectures.
>
> **W1-2. A step-by-step explanation of how the weight values affect robustness accuracy**
>
> **A.** Using Lemmas 1 and 2, we demonstrated that in the optimal case, the weight values $w_1, w_2, \ldots, w_k$ are equal. Under this condition, the standard and robustness accuracies are independent of the specific weight values. Instead, the accuracies are determined solely by the ratio between the weights and the bias term. This is the key reason we applied $ l_2 $-normalization for simplicity, as shown in the main paper. Below, we provide detailed step-by-step proof to clarify this relationship.
> By Lemma 1, the optimal classifier satisfies $w_1 = w_2 = \cdots = w_n = w$.
> From Definition 1, the standard error is defined as:
> $$\mathcal{R}\_\text{std}(f) = \Pr(f(\boldsymbol{x}) \neq y),$$$$= \Pr (y=+1)\cdot\mathcal{R}^{+1}\_{nat}(f_{nat}) + \Pr (y=-1)\cdot\mathcal{R}^{-1}\_{nat}(f_{nat})$$$$
> = \frac{r}{r+1} \Pr \left(f(\boldsymbol{x}) \neq y \mid y = +1\right) + \frac{1}{r+1}\Pr \left(f(\boldsymbol{x}) \neq y \mid y = -1\right)
> $$$$
> = \frac{r}{r+1} \Pr \left(\sum^n_{k=1} w \mathcal{N}(+\eta, 1) + b  < 0\right) + \frac{1}{r+1}\Pr \left(\sum^n_{k=1} w \mathcal{N}(-\eta, 1) + b  > 0\right)
> $$$$
> = \frac{r}{r+1} \Pr \left( \mathcal{N}(+n w \eta, n w^2) + b  < 0\right) + \frac{1}{r+1}\Pr \left( w \mathcal{N}(-n w \eta, n w^2) + b  > 0\right)
> $$$$
> = \frac{r}{r+1} \Pr \left( \mathcal{N}(0, 1)   < \frac{-nw\eta - b}{\sqrt{nw^2}}\right) + \frac{1}{r+1}\Pr \left(\mathcal{N}(0, 1)  < \frac{-nw\eta + b}{\sqrt{nw^2}}\right)
> $$$$
> = \frac{r}{r+1}\cdot\Phi\left(\frac{-nw\eta - b}{\sqrt{nw^2}} \right) +  \frac{1}{r+1}\cdot\Phi\left(\frac{-nw\eta + b}{\sqrt{nw^2}}\right)
> $$Here, $\Phi$ is the cumulative distribution function of the standard normal distribution.$$
> \frac{\partial \mathcal{R}\_{nat}(f\_{nat})}{\partial b} = -\frac{r}{r+1}\cdot\phi\left(\frac{-nw\eta - b}{\sqrt{nw^2}} \right) +  \frac{1}{r+1}\cdot\phi\left(\frac{-nw\eta + b}{\sqrt{nw^2}} \right) = 0$$$$
> -r\cdot\phi\left(\frac{-nw\eta - b}{\sqrt{nw^2}} \right) +  \phi\left(\frac{-nw\eta + b}{\sqrt{nw^2}} \right) = 0$$
> Here, $\phi$ is the standard normal distribution function.
>
> $$
> \frac{\partial \mathcal{R}\_{nat}(f\_{nat})}{\partial w} = \frac{r}{r+1}\cdot\left(\frac{b}{ w\sqrt{nw^2}}\right)\cdot\phi\left(\frac{-nw\eta - b}{\sqrt{nw^2}} \right) +  \frac{1}{r+1}\cdot\left(\frac{-b}{ w\sqrt{nw^2}}\right)\cdot\phi\left(\frac{-nw\eta + b}{\sqrt{nw^2}} \right) = 0
> $$
>
> $$
> r\cdot\phi\left(\frac{-nw\eta - b}{\sqrt{nw^2}} \right) -  \phi\left(\frac{-nw\eta + b}{\sqrt{nw^2}} \right) = 0
> $$
> Note that the two conditions are equivalent.$$
> -r\cdot\exp\left(-\frac{1}{2}\left(\frac{-n\eta - b}{\sqrt{nw^2}} \right)^2\right) +  \exp\left(-\frac{1}{2}\left(\frac{-nw\eta + b}{\sqrt{nw^2}} \right)^2\right) = 0$$$$
> \ln r -\frac{1}{2}\left(\frac{-nw\eta - b}{\sqrt{nw^2}} \right)^2 =  -\frac{1}{2}\left(\frac{-nw\eta + b}{\sqrt{nw^2}} \right)^2$$
>
> Therefore, the optimal ratio $b^*\_{nat}$ to $w^*\_{nat}$ of natural classifier is:$$
> \frac{b^*\_{nat}}{w^*\_{nat}} = \frac{\ln r}{2\eta}.$$
> Then, the natural and robust error for the tail class is as follows. (For robust error, using Equation (13))$$
> \mathcal{R}\_{nat}(f\_{nat} | -1) = \Phi\left(\frac{-nw^*\_{nat}\eta + b^*\_{nat}}{\sqrt{n(w^*\_{nat})^2}}\right) = \Phi\left(-\sqrt{n}\eta + \frac{\ln r}{2\sqrt{n}\eta}\right)$$$$
> \mathcal{R}\_{rob}(f\_{nat} | -1) = \Pr(\exists \boldsymbol{\delta} \text{ with } \| \boldsymbol{\delta} \|\_\infty \leq \epsilon \text{ s.t. } f^*_{nat}(\boldsymbol{x} + \boldsymbol{\delta}) \neq y | y=-1) = \Phi\left(\frac{-nw^*\_{nat}(\eta-\epsilon) + b^*\_{nat}}{\sqrt{n(w^*\_{nat})^2}}\right) = \Phi\left(-\sqrt{n}(\eta-\epsilon) + \frac{\ln r}{2\sqrt{n}\eta}\right)
> $$Similarly, the optimal robust classifier can be obtained based on Lemma 2, with the only difference being $\eta-\epsilon$ instead of $\eta$.

---

> > ### Author Response · Authors · 2024-11-19
> > **Reply to Reviewer Uh2k. (2/2)**
> >
> > **W2. Connection Between Figure 2 and Theorem 1**
> >
> > **A.**
> > Figure 2 provides a visualization of the Equation (2) within a 2-dimensional space. The data were sampled from Gaussian distributions with $\eta = 1$ and $n = 2$, following Equation (2), considering three different imbalance ratios (IR = 1, 2, 5).
> > In this figure, the black line represents the ideal decision boundary, the blue line corresponds to the decision boundary produced by the natural classifier, and the red line represents the decision boundary produced by the adversarially trained classifier.
> >
> > When IR = 1 (i.e., the number of samples for the +1 and -1 classes are equal), both classifiers—the natural and robust classifiers— perfectly align with the ideal decision boundary, as shown in the first plot. However, in the second and third plots, as IR increases (i.e., the number of samples in the -1 class, shown in green, becomes fewer than those in the +1 class, shown in yellow), both classifiers deviate further from the ideal decision boundary. Notably, the robust classifier (red) deviates more significantly than the natural classifier (blue) from the ideal boundary.
> >
> > This observation aligns with Corollary 1, which states that adversarial training on long-tailed datasets exacerbates the vulnerability of the tail class, making them even less robust than under natural training. As the robust classifier deviates further from the ideal decision boundary and becomes more skewed toward the side with more tail-class samples, it exhibits higher natural and robust errors for the tail class compared to the natural classifier. Moreover, as described in Theorem 1, all tail-class error increases as a function of the imbalance ratio $r$. This is evident in the comparison between IR = 2 and IR = 5, where both the natural and robust classifiers exhibit a stronger bias toward the tail class as IR increases.
> >
> > A quantitative evaluation of these observations is provided in Figure 3. This figure plots the accuracies (the complement of the errors) derived from the scenarios depicted in Figure 2. The robust accuracy clearly shows that the robust classifier achieves lower accuracy than the natural classifier in these imbalanced settings, reflecting its higher natural and robust errors.
> >
> > **W3. Definition of Variables in Equations 4 and 5**
> >
> > **A.**
> > The definition of $r$, the imbalance ratio, is provided in the main paper: $ r \geq 1 $, i.e., $r $ represents the ratio of the number of instances in the head class ($ y = +1 $) to the number of instances in the tail class ($ y = -1 $). Additionally, $\Phi $ is the cumulative distribution function of the standard normal distribution. To improve clarity and readability, we will explicitly reference these definitions in the main paper in the revised version.
> >
> > **W4. Code Availability**
> >
> > **A.** We apologize for not explicitly mentioning this in the main paper. Please note that the full code has already been provided in the supplementary material for complete transparency and reproducibility.
> >
> > **Weakness in Summary part: The proof of Corollary indicates that it is trivial according to equations 4 and 5, which appears to indicate that the readers should figure it out themselves.**
> >
> > **A.** We believe this misunderstanding arose because we did not explicitly define $\Phi$ as the cumulative distribution function of the standard normal distribution in the main text. To clarify, $\Phi$ is an increasing function, and all variables in Theorem 1 (particularly $\eta$ and $\epsilon$) are positive. Therefore, it is straightforward to deduce that $\mathcal{R}^{-1}\_{rob}(f^*\_{rob}) > \mathcal{R}^{-1}\_{rob}(f^*\_{nat})$
> >
> >
> > **Q1. Dataset for Figure 3**
> >
> > **A.** We used the same binary dataset as depicted in Figure 2 for Figure 3. The data were sampled from Gaussian distributions with $\eta = 1$ and $n = 2$, following Equation (2), considering three different imbalance ratios (IR = 1, 2, 5).
> >
> > **Q2. Comparison with Balanced Datasets**
> >
> > **A.** In Figure 3, IR=1 represents a balanced dataset, where the +1 class and -1 class have the same number of samples.
> >
> > **Q3. IR = 1 Results in Figure 3**
> >
> > **A.** Both natural and adversarial training align with the ideal decision boundary, as there is no bias. This can be directly derived from the results of Theorem 1.
> >
> > **Q4. Loss Function in Algorithm 1**
> >
> > **A.** The loss function used in Algorithm 1 is IGDM loss from the adversarial distillation method [1]. Formulaic representation is
> > $
> > \mathcal{L}_{KD}(f(\boldsymbol{x}'_i), f_b(\boldsymbol{x}_i)) = \text{KL}(f(\boldsymbol{x}'_i) - f(\boldsymbol{x}_i) \|\| f_b(\boldsymbol{x}'_i) - f_b(\boldsymbol{x}_i))
> > $, where $\text{KL}$ is KL divergence loss.
> >
> > [1] Hongsin Lee, Seungju Cho, and Changick Kim. Indirect gradient matching for adversarial robust distillation. arXiv preprint arXiv:2312.03286, 2023.

---

> > > ### Author Response · Authors · 2024-11-21
> > > **Dear reviewer Uh2k.**
> > >
> > > Thank you for reviewing our submission and sharing your valuable feedback.
> > >
> > > We have carefully addressed your comments and provided detailed responses. We hope our explanations resolve the concerns you raised. If there are any further questions or points requiring clarification, please feel free to let us know. We truly appreciate your time and effort in reviewing our work.
> > >
> > > Best regards, Authors of paper number 5341.

---

> > > > ### Author Response · Authors · 2024-11-25
> > > > **Dear reviewer Uh2k.**
> > > >
> > > > We are reaching out as the discussion period is ending soon to kindly remind you about our responses to your review.
> > > >
> > > > We have carefully addressed your comments and hope they resolve your concerns.
> > > >
> > > > If there’s anything further you’d like us to clarify, please let us know before the deadline.
> > > >
> > > > Thank you again for your time and feedback.

---

> > > > > ### Author Response · Authors · 2024-11-29
> > > > > **Final Reminder to Dear Reviewer Uh2k.**
> > > > >
> > > > > As the discussion period is coming to a close, we would like to kindly remind you of our detailed responses to your insightful review comments. We have thoroughly addressed each of your points and believe our clarifications resolve the concerns you raised.
> > > > >
> > > > > If there are any remaining questions or points that require further elaboration, we would be happy to provide additional information. Please let us know if there’s anything else we can do to assist before the deadline.
> > > > >
> > > > > Thank you once again for your consideration and support.

---

> > ### Comment · Reviewer_Uh2k · 2024-11-29
> >
> > W1-1. The question was more about whether the proof given is applicable to the general learning process, such as the discrepancy between weight modification leading to a discrepancy in performances is a general case. So, how is the given proof specific to long-tailed cases?
> >
> > W1-2.  is not readable.

---

> > > ### Comment · Reviewer_Uh2k · 2024-11-29
> > >
> > > I believe authors have answer/clarify most of my question. I will keep my rating as it is.

---

> > > > ### Author Response · Authors · 2024-12-02
> > > > **Reply to reviewer Uh2k.**
> > > >
> > > > Thank you for taking the time to review our paper and for acknowledging our clarifications. We appreciate your constructive feedback throughout the process.

---

### Official Review · Reviewer_ffZU · 2024-11-03

**Soundness:** 3
**Presentation:** 4
**Contribution:** 3
**Rating:** 8
**Confidence:** 3

**Summary:**

This paper introduces a method to improve adversarial robustness on long-tailed distributions. The primary contributions include a two step adversarial training paradigm which involve, (i) adversarially training a teacher model on a balanced sub-dataset obtained from the original unbalanced dataset, and (ii) adversarially training a student model on the unbalanced original dataset using knowledge distillation from the teacher model and an additional balanced softmax loss. Another notable contribution is a theoretical analysis demonstrating how traditional adversarial training methods fail to improve robust accuracy of tail classes even when compared to non-adversarially trained methods.

**Strengths:**

- The authors effectively justify the need for adversarial training methods tailored to long-tailed distributions through their theoretical analysis (Section 3.2) and empirical evaluations (Section 3.3).

- An intriguing result is the observation that adversarial training unexpectedly reduces robust performance on tail classes compared to non-adversarial methods when addressing long-tailed distributions.

- The experimental results demonstrate substantial improvements in both clean and robust accuracy on tail classes, outperforming existing adversarial training (AT) methods such as TRADES, MART, and AWP, as well as long-tailed AT methods like RoBal, REAT, and AT-BSL.

- These performance gains are consistently observed across multiple datasets and architectural frameworks, highlighting the method's potential usability.

**Weaknesses:**

- The paper does not sufficiently explain why their proposed method outperforms existing AT methods focused on long-tailed distributions. Particularly, why should Knowledge Distillation be a good choice for handling adversarial-robustness on long-tailed datasets?

- The incorporation of knowledge distillation (KD) and balanced softmax loss appears somewhat arbitrary. E.g., KD has been already explored for training of long-tailed datasets [1,2,3]; balanced softmax has been used in AT for long-tailed data [4,5]; KD for AT is explored in [6]. However, I agree that KD is not used for all AT and long-tailed datasets simultaneously. Hence, I would like to get a better justification of the proposed method for clarity.

- The absence of a dedicated subsection that develops a theoretical or intuitive motivation for the proposed method diminishes the overall impact and comprehensibility of the paper. Providing a more thorough theoretical framework or intuitive explanations would significantly strengthen the paper’s contributions and clarity.

- The proof of Lemma 1 in Appendix A needs to be a bit more rigorous and show how having equal weights minimizes the variance of $\sum_{k=1}^n w_k x_k + b$ using Cauchy-Schwartz, and hence minimises the natural error.

- Some inconsistency in the naming of the balanced dataset, throughout the paper, but specifically on page 6, line 317 onwards, notation shifts from D_b to D_B.




Overall, the paper presents strong experimental results and provides sound theoretical analysis to motivate the original problem statement—adversarial training for long-tailed distributions. Therefore, I provide a score of 6 (Weak Accept). If the authors address the concerns outlined in the Weakness section, I would be willing to raise my score.



[1] T. Li, L. Wang, and G. Wu, “Self supervision to distillation for long-tailed visual recognition,” in IEEE International Conference on Computer Vision, 2021, pp. 630–639.

[2] Y.-Y. He, J. Wu, and X.-S. Wei, “Distilling virtual examples for long-tailed recognition,” in IEEE International Conference on Computer Vision, 2021, pp. 235–244.

[3] H. Rangwani, P. Mondal, M. Mishra, A. R. Asokan, and R. V. Babu, “Deit-lt: Distillation strikes back for vision transformer training on long-tailed datasets,” in IEEE Conference on Computer Vision and Pattern Recognition (CVPR), 2024, pp. 23396–23406.

[4] TongWu, Ziwei Liu, Qingqiu Huang, YuWang, and Dahua Lin. Adversarial robustness under longtailed distribution. In Proceedings of the IEEE/CVF Conference on Computer Vision and Pattern Recognition (CVPR), pp. 8659–8668, June 2021.

[5] Xinli Yue, Ningping Mou, Qian Wang, and Lingchen Zhao. Revisiting adversarial training under long-tailed distributions. In Proceedings of the IEEE/CVF Conference on Computer Vision and Pattern Recognition (CVPR), pp. 24492–24501, June 2024.

[6] Hongsin Lee, Seungju Cho, and Changick Kim. Indirect gradient matching for adversarial robust distillation. arXiv preprint arXiv:2312.03286, 2023.


--------------------

**Post Rebuttal**:


Thank you for addressing my concerns and providing additional clarity in the rebuttal. I appreciate the authors' efforts to strengthen the motivation for their method, particularly with respect to the usage of KD alongside the Balanced Softmax loss.

> ⁠Consequently, we argue that if the teacher is robust to the tail classes, then using KD enables the student model to mimic the teacher’s robustness and perform similarly well in the tail classes. There are various ways to make the teacher robust to tail classes, but in our case, we provided theoretical proof showing that a teacher trained on a balanced dataset is more robust to tail classes compared to one trained on an unbalanced dataset.

I am satisfied with the justification provided for the approach, as well as the addition of these explanations to Sections 4.1 and 4.2 of the paper. This effectively addresses my primary concern about the motivation behind the method.

As a result, I am increasing my score.

**Questions:**

See weaknesses.

---

> ### Author Response · Authors · 2024-11-19
> **Reply to Reviewer ffZU.**
>
> Thank you for your thoughtful and supportive review. We appreciate your recognition of our theoretical and empirical contributions and the performance gains of our method.
>
> **W1. Why should Knowledge Distillation be a good choice?**
>
> **A.** We would like to emphasize that our use of  Knowledge Distillation (KD) is not to claim that it inherently addresses long-tailed distributions effectively but rather to highlight its role in effectively transferring the teacher’s knowledge on the tail’s robustness. Specifically, in the field of adversarial distillation, it is well established that KD can successfully replicate the robustness of the teacher, as shown in prior works [1], [2]. Consequently, we argue that if the teacher is robust to the tail classes, then using KD enables the student model to mimic the teacher’s robustness and perform similarly well in the tail classes. There are various ways to make the teacher robust to tail classes, but in our case, we provided theoretical proof showing that a teacher trained on a balanced dataset is more robust to tail classes compared to one trained on an unbalanced dataset. This theoretical insight is presented in Theorem 1 and Corollary 1, and the teacher’s performance on tail classes is detailed in Table 6.
> In summary, we first trained the teacher to be robust to tail classes using a simple yet effective approach. Then, we leveraged KD to ensure that this robustness to tail classes is well transferred.
>
> [1] Huang, Bo, et al. "Boosting accuracy and robustness of student models via adaptive adversarial distillation." Proceedings of the IEEE/CVF Conference on Computer Vision and Pattern Recognition. 2023.
> [2] Hongsin Lee, Seungju Cho, and Changick Kim. Indirect gradient matching for adversarial robust distillation. arXiv preprint arXiv:2312.03286, 2023.
>
> **W2. Justification for Combining Knowledge Distillation and Balanced Softmax**
>
> **A.**  Balanced Softmax is a powerful method, as demonstrated extensively in AT-BSL and RoBal, where it effectively addresses the issue of tail-class robustness under adversarial training. These works show that merely applying Balanced Softmax significantly improves the robustness of tail classes, preventing their robustness from collapsing to zero. However, our primary motivation stems from the observation that Balanced Softmax alone is insufficient to guarantee robustness for tail classes.
> To address this limitation, we incorporated Knowledge Distillation (KD) as a complementary mechanism. KD ensures that the robustness of the self-teacher model, particularly for the tail classes, is effectively transferred to the student model.
> It is important to acknowledge that using KD alone, without Balanced Softmax, would essentially replicate existing adversarial distillation methods. This approach, however, suffers from a critical drawback: the use of non-robust self-teacher results in a significant performance drop, particularly for head classes. By combining Balanced Softmax with KD, we leverage the strengths of both techniques—Balanced Softmax mitigates class imbalance effectively, while KD further enhances the robustness of tail classes—resulting in a more balanced and robust overall performance.
>
> **W3. Addressing the Need for Clearer Motivation for the proposed method**
>
> Thank you for pointing out this valuable suggestion. In response, we have revised the paper to better address the intuitive motivations for our proposed method. These improvements have been incorporated into Section 4, highlighted in blue, to enhance the paper's clarity and impact.
>
> **W4. Clarification and Rigor in the Proof of Lemma 1**
>
> While the use of the Cauchy-Schwarz inequality is a possible approach, since the data distribution is modeled as i.i.d. Gaussian variables, the variance of the linear combination can be directly calculated as follows.
>
>  Given the data distribution, we suppose $x_i$ follows Gaussian distribution as follows.
>
> $$ x_1, \dots, x_n \overset{i.i.d}{\sim} \mathcal{N}(\eta y, 1) $$ The variance of $ \sum_{k = 1}^{n} w_k x_k + b $ can be calculated as follows: $$ \text{Var}\left(\sum_{k = 1}^{n} w_k x_k + b\right) = \text{Var}\left(\sum_{i=1}^n w_i x_i\right) = \sum_{i=1}^n w^2\_i$$ Note that $$ \sum_{i=1}^n w_i x_i \sim \mathcal{N}\left(\sum_{i=1}^n w_i \eta y, \sum_{i=1}^n w^2\_i \right). $$
>
> We demonstrated that optimal weights are equal through a proof by contradiction. Assuming that optimal weights exist where the weights are not equal leads to a contradiction. Therefore, by proof by contradiction, the optimal weights must be equal. This is formally shown in Lemma 1 of the appendix. The specific expression for the natural error is provided in Equation (9) in the appendix.
>
>
> **W5. Some inconsistency in the naming of the balanced dataset, throughout the paper, but specifically on page 6, line 317 onwards, notation shifts from D_b to D_B.**
>
> **A.** Thanks for your comments. We have updated our paper.

---

> > ### Author Response · Authors · 2024-11-21
> > **Dear reviewer ffZU.**
> >
> > Thank you for reviewing our submission and sharing your valuable feedback.
> >
> > We have carefully addressed your comments and provided detailed responses. We hope our explanations resolve the concerns you raised. If there are any further questions or points requiring clarification, please feel free to let us know. We truly appreciate your time and effort in reviewing our work.
> >
> > Best regards, Authors of paper number 5341.

---

> > > ### Comment · Reviewer_ffZU · 2024-11-21
> > >
> > > Thanks for your clarifications. I have increased my score.

---

> > > > ### Author Response · Authors · 2024-11-22
> > > > **Reply to Reviewer ffZU.**
> > > >
> > > > Thank you for taking the time to review our responses and for your thoughtful reconsideration.
> > > > We sincerely appreciate your feedback and are grateful that our clarifications could address your concerns.

---

### Official Review · Reviewer_pWdA · 2024-11-03

**Soundness:** 3
**Presentation:** 2
**Contribution:** 2
**Rating:** 8
**Confidence:** 4

**Summary:**

In this paper, the authors first provide a theoretical analysis explaining why adversarial training leads to robustness degradation in tail classes. Motivated by these findings, they propose a simple empirical solution to mitigate this issue. Specifically, they employ a distillation-based approach in which a balanced version of the long-tailed dataset is used to adversarially train a teacher model. This teacher model is then used to train a student model on the original long-tailed dataset. The authors conduct experiments across multiple datasets and various settings, such as with and without augmentations, different imbalance ratios, and more.

**Strengths:**

The key strengths of the paper are:
- The paper is mostly written well.
- The authors conduct comprehensive experiments, as well as provide theoretical motivations.
- While the improvement methods use standard techniques like balanced softmax and indirect gradient matching, it's interesting to see that a simple prior distillation step enhances performance.

**Weaknesses:**

- The concept of Imbalance Ratio in the paper is not well explained. While I understand how it applies to a binary setup, it’s unclear what an imbalance ratio of 50 means for CIFAR-10-LT compared to an imbalance ratio of 10. I suggest that the authors include the number of samples in the least populated class for clarity.
- Related to this, the core assumption is that training on a balanced dataset will enhance feature learning for the tail classes. However, the proposed method may negatively impact overall performance if the long-tail ratio is extreme (e.g., 5 samples per class). This could lead to ineffective training of the teacher network, and the distillation loss may then degrade performance. I would be interested to hear the authors' thoughts on this.
- Additionally, given the smaller sample size (especially for tail classes), it would be helpful to report the mean along with error metrics, such as standard deviation, and to conduct statistical tests to verify that the proposed method significantly improves performance compared to other methods.

**Questions:**

- Please see weaknesses and clarify the imbalance ratio better, and conduct statistical tests for comparison with other methods.
- It’s unclear why the authors report the AutoAttack accuracy on the whole dataset but not for the tailed classes. I would like to see the AutoAttack accuracy on the tailed classes for Figure 1 and all relevant tables.

---

> ### Author Response · Authors · 2024-11-19
> **Reply to Reviewer pWdA.**
>
> We appreciate the reviewer’s recognition of our work, particularly highlighting the comprehensive experiments and the theoretical motivations that underpin our approach.
>
> **W1. The concept of Imbalance Ratio in the paper is not well explained.**
> **A.** We apologize for the lack of clarity. The number of samples per class in an imbalanced dataset is determined by the imbalance ratio (IR) using the following exponential formula:
>
> $n_i = N_i * {\left(\frac{1}{IR}\right)^{\frac{i}{C-1}}} $
>
> , where $N_i$ is the original number each class, $C$ is the number of total classes, and $i$ is the class index.
>
> For CIFAR-10-LT, the total number of training samples in each class is 5000, and the number of classes $C$ is 10. Below are the class sample sizes for different IR values:
>
> | i   | IR = 10 | IR = 50 | IR = 100 |
> |-----|---------|---------|----------|
> | 0   | 5000    | 5000    | 5000     |
> | 1   | 3871    | 3237    | 2997     |
> | 2   | 2997    | 2096    | 1796     |
> | 3   | 2320    | 1357    | 1077     |
> | 4   | 1796    | 878     | 645      |
> | 5   | 1391    | 568     | 387      |
> | 6   | 1077    | 368     | 232      |
> | 7   | 834     | 238     | 139      |
> | 8   | 645     | 154     | 83       |
> | 9   | 500     | 100     | 50       |
>
>
>
> **W2. The proposed method may negatively impact overall performance if the long-tail ratio is extreme.**
>
> **A.** We agree that if the amount of data decreases significantly, the performance of the teacher will deteriorate. Note that in such an extreme case, the overall performance of all the baselines will also decrease. To emphasize the superiority of our method, we present experimental results (see the reply below) for an imbalance ratio of 100 in ~~CIFAR-10-LT~~ CIFAR-100-LT, where the tail class contains only 5 samples. Thank you for your comments.
>
>
>
>
> **W3. Report the mean, standard deviation compared to other methods given the smaller sample size (especially for tail classes)**
>
> **A.**
>
> The clean accuracy and robustness with 100 of IR using ResNet-18 on CIFAR-10-LT. T-Clean and T-PGD are clean and 20-step of PGD accuracy on the tail class group.
> | **Method**      | **Clean**            | **PGD**              | **AA**               | **T-Clean**         | **T-PGD**          |
> |------------------|----------------------|----------------------|----------------------|---------------------|---------------------|
> | RoBal           | 66.73 ± 0.26        | 22.64 ± 0.28        | 20.34 ± 0.35        | 52.44 ± 2.53       | 6.3 ± 1.09         |
> | Reat            | 64.84 ± 0.52        | 25.93 ± 0.45        | 23.83 ± 0.22        | 50.08 ± 3.43       | 6.3 ± 0.98         |
> | BSL             | 64.05 ± 0.33        | 26.30 ± 0.38        | 24.21 ± 0.34        | 45.02 ± 3.02       | 5.5 ± 1.09         |
> | **Ours**        | **66.10 ± 0.54**    | **30.24 ± 0.34**    | **26.76 ± 0.31**    | **53.04 ± 2.14**   | **12.12 ± 0.97**   |
>
>
>
> The clean accuracy and robustness with 100 of IR using ResNet-18 on CIFAR-100-LT. T-Clean and T-PGD are clean and 20-step of PGD accuracy on the tail class group.
>
> | Method       | Clean             | PGD               | AA                | T-Clean           | T-PGD             |
> |--------------|-------------------|-------------------|-------------------|-------------------|-------------------|
> | RoBal        | 31.22 ± 0.22     | 10.63 ± 0.21     | 9.68 ± 0.17      | 0.28 ± 0.15      | 0.08 ± 0.06      |
> | Reat         | 32.64 ± 0.55     | 10.81 ± 0.24     | 9.81 ± 0.11      | 10.80 ± 0.62     | 1.18 ± 0.08      |
> | BSL          | 33.20 ± 0.34     | 10.66 ± 0.08     | 9.78 ± 0.10      | 10.60 ± 0.60     | 1.22 ± 0.27      |
> | **Ours**     | **34.44 ± 0.21** | **14.63 ± 0.18** | **12.32 ± 0.08** | **11.21 ± 0.61** | **2.60 ± 0.15** |
>
> These tables represent the mean values calculated from five different seeds, and the values after ± indicate the standard deviations. Our method outperformed the baselines even when the imbalance ratio (IR) was 100. However, improving performance in such extreme cases with very limited data remains a challenge.
>
>
> **Q2. AutoAttack accuracy for tail classes.**
>
>
> **A.** We report the AutoAttack accuracy on the tail classes (T-AA) as follows. Thanks for your comments.
>
> | | CIFAR-10-LT || CIFAR-100-LT || Tiny-ImageNet-LT ||
> |-|-|-|-|-|-|-|
> | **Method**     | **T-PGD**       | **T-AA** | **T-PGD**        | **T-AA** | **T-PGD**            |**T-AA**|
> | PGD-AT     | 2.2    | 0.0  | 6.7      | 5.9 | 5.0  | 3.8 |
> | TRADES     | 0.6      | 0.0  | 6.8        | 6.1 | 6.0     | 4.5 |
> | MART       | 1.0    | 0.0  | 7.2      | 6.1 | 7.6       | 4.9 |
> | AWP        | 0.6         | 0.0  | 8.6      | 7.1 | 6.6       | 4.8 |
> | RoBal      | 15.9        | 13.5| 6.7          | 6.4 | 4.8         | 3.2 |
> | REAT       | 11.6        | 11.2| 10.5         | 9.0 | 8.8         | 5.6 |
> | AT-BSL     | 8.7         | 6.9 | 9.1          | 6.9 | 7.0        | 6.4 |
> | **Ours**   | **36.2**        | **32.5**| **17.6**         | **13.2**| **12.6**             | **9.4** |

---

> > ### Author Response · Authors · 2024-11-21
> > **Dear reviewer pWdA.**
> >
> > Thank you for reviewing our submission and sharing your valuable feedback.
> >
> > We have carefully addressed your comments and provided detailed responses. We hope our explanations resolve the concerns you raised. If there are any further questions or points requiring clarification, please feel free to let us know. We truly appreciate your time and effort in reviewing our work.
> >
> > Best regards,
> > Authors of paper number 5341.

---

> > ### Comment · Reviewer_pWdA · 2024-11-24
> >
> > Thanks for your response.
> >
> > Overall, I am satisfied with the author's response regarding experiments as IR increases and I am increasing my score.
> >
> > > imbalance ratio of 100 in CIFAR-10-LT, where the tail class contains only 5 samples
> >
> > As per the table and formula, shouldn't it be 50?
> >
> >  For the CR, I think it would make the paper stronger by having more ablations at different IR values. It would be interesting to see where the gap between different methods starts converging.

---

> > > ### Author Response · Authors · 2024-11-24
> > > **Reply to Reviewer pWdA.**
> > >
> > > Thank you for your thoughtful response and for considering our clarifications. I sincerely appreciate your insights and suggestions, which have been invaluable in strengthening our work.
> > >
> > > Regarding the imbalance ratio (IR), we apologize for the typo in our earlier response. The correct reference is to CIFAR-100-LT, where the tail class contains 5 samples when IR is 100. Thank you for catching this oversight.
> > >
> > > We also sincerely appreciate your suggestion about conducting more ablations at different IR values. It’s an insightful idea that aligns with our interest in understanding where performance gaps between methods converge. In our initial experiments with extreme values like IR=1000, we observed that tail-class robustness becomes negligible, with all methods converging to near-zero performance. We plan to expand our analysis to include a broader range of IR values and further explore methods that can achieve strong robustness even under such extreme IR conditions in future work.
> > >
> > > Thank you again for your constructive comments and support!

---

### Official Review · Reviewer_61WA · 2024-11-08

**Soundness:** 3
**Presentation:** 3
**Contribution:** 3
**Rating:** 6
**Confidence:** 4

**Summary:**

This paper improves model robustness against adversarial attacks in long-tailed learning.

The authors observed that adversarial training, compared with normal training, further sacrifices minority class accuracy over majority classes. This observation is interesting and quite intuitive after some thoughts. It could be inspiring to further works.

Based on this observation, the authors proposed to use stronger constrains than previous methods to balance the accuracy among different classes. Specifically, previous methods uses balance-aware loss functions to balance accuracy in long-tailed adversarial training. The authors added resampling (when training the teacher model), another common technique to handle data imbalance, on top of those previous works. The overall method is limited in technical novelty.

Experimental results show good improvements over previous methods. The experimental settings raise some concerns to me.

Overall, I think this is a marginal paper. I lean toward accepting primarily due to the good numerical results shown under the specific experiment setting used by this paper.

**Strengths:**

1. The observation that adversarial training further harms minority classes compared to normal training is interesting and may inspire further works.

2. The performance gain under the specific experiment setting is good.

**Weaknesses:**

1. The technical novelty is limited. The two methods used to handle long-tail learning in this paper are resampling (when training the teacher model) and a balance-aware loss function proposed by previous works.

2. The proposed method has a noticeable limitation: It requires to train a teacher model on balanced data. This greatly limits its practical application. In most cases, the powerful large model used as teacher models (e.g., foundation models downloaded from Hugging faces) are trained on real-world datasets which are typically unbalanced. It is very hard to get an off-the-shelf model that is "balanced". This requires the users to train the balanced model on their own. As a training algorithm, this is considerable complexity overhead. Moreover, it is not discussed in the paper how the teacher model could affect the distillation. Does the re-sampling method matter? Does the size of the teach model matter? Does the re-sampled dataset has to be perfectly balanced or it could have some imbalance? If it is the later case, how much imbalance it could have?

3. Table 5 is not interpretable. Which part is CIFAR10 and which part is CIFAR100?

4. The experiment setting is limited. The largest imbalance ratio is 50 for CIFAR10 and 10 for CIFAR100 and ImageNet. However, on both datasets, it is common to experiment with much stronger imbalance, with the ratio up to 100 [1]. Experimenting with stronger imbalance ratio is important since it is a more challenging case for re-sampling based methods. Under the stronger ratio, I assume the training of the teacher model would face difficulty cause the minority classes have few samples and simple up-sampling might not help too much.

[1] Learning Imbalanced Datasets with Label-Distribution-Aware Margin Loss.

**Questions:**

Please see above.

---

> ### Author Response · Authors · 2024-11-19
> **Reply to Reviewer 61WA. (1/3)**
>
> We sincerely appreciate your time and effort in reviewing our work. Your valuable insights and constructive feedback are helpful in refining and strengthening our research.
>
> **W1. The technical novelty is limited. The two methods used to handle long-tail learning in this paper are resampling (when training the teacher model) and a balance-aware loss function proposed by previous works.**
>
> **A.** Our novelty comes from a self-distillation framework. Sampling and balance aware loss is a kind of fundamental loss for long-tailed distribution.
>
> By theoretically analyzing the limitations of conventional long-tail adversarial training approaches, we identified a gap in robustness for tail classes that needed to be sufficiently addressed. Through our integration of balanced sampling and novel self-distillation framework, we aim to provide a practical solution to improve tail-class robustness, which we believe is a crucial step toward advancing adversarial robustness in imbalanced settings.

---

> ### Author Response · Authors · 2024-11-19
> **Reply to Reviewer 61WA. (2/3)**
>
> **W2. Response to Concern Regarding the Practicality of Training a Balanced Teacher Model.**
>
> **A.** Thank you for raising this concern about the requirement to train a teacher model on balanced data. We would like to address this issue by clarifying key points and contextualizing our method within the broader landscape of adversarial training and long-tail learning.
>
> 1) Lightweight Self-Teacher Design:
>
> The teacher model in our method is not intended to be a powerful or large foundation model but rather a **lightweight self-teacher** trained on a balanced sub-dataset constructed from the **given unbalanced dataset**. This self-teacher is explicitly designed to improve tail-class robustness with minimal computational overhead. If a long-tail dataset (e.g., ImageNet-LT, CIFAR-100-LT) is available, balanced sampling can be easily performed by re-sampling or over-sampling underrepresented classes. In our experiments, we demonstrated the effectiveness of this approach by training the lightweight self-teacher on balanced datasets using only a small number of epochs (30). While the overall performance of the self-teacher might insufficient due to the small number of training data, it achieves balanced robustness across all classes, particularly for the tail class. Importantly, this self-teacher requires only a small number of epochs (30)  to train, making it computationally efficient and practical.
>
> 2) Lack of Adversarially Trained Models for Long-Tail Datasets:
>
> Currently, there are no off-the-shelf adversarially trained models specifically designed for long-tail datasets, such as ImageNet-LT, or even CIFAR-10/100-LT. Publicly available adversarially trained models, such as those listed on RobustBench, are almost exclusively trained on balanced datasets like CIFAR-10, CIFAR-100, and ImageNet. These models are highly robust but are not designed for long-tail datasets. This highlights a broader limitation in adversarial training research rather than a specific drawback of our method. In fact, the absence of adversarially trained models for long-tail scenarios underscores the necessity and relevance of our approach.
>
> 3) Availability of Highly-robust Teacher model:
>
> We appreciate the suggestion regarding the potential use of highly robust teacher models. Indeed, employing a robust teacher trained on a balanced dataset can lead to significant performance improvements. To evaluate this scenario, we conducted additional experiments using pre-trained adversarially robust teachers (e.g., models from RobustBench) to perform distillation on long-tail datasets. The results show that even with such pre-trained robust teachers(from Robustbench), the proposed method effectively improves robustness for long-tail datasets.
> | **Method**                              | **Clean** | **PGD** | **AA**  | **T-Clean** | **T-PGD** |
> |-----------------------------------------|-----------|----------|----------|-------------|-----------|
> | **RoBal**                               | 37.97     | 10.51    | 8.64     | 21.8        | 4.8       |
> | **REAT**                                | 38.48     | 10.58    | 9.07     | 31.6        | 8.8       |
> | **AT-BSL**                              | 38.41     | 10.25    | 8.90     | 32.4        | 7.0       |
> | **Ours with balanced Self-teacher (Proposed)** | 49.37     | 14.09    | 11.15    | 33.8        | 12.6      |
> | **Ours with teacher [1]**               | 57.03     | 30.18    | 24.79    | 46.3        | 24.2      |
> | **Ours with teacher [2]**               | 50.98     | 28.13    | 22.93    | 39.3        | 22.0      |
> | **Ours with teacher [3]**               | 55.95     | 28.39    | 22.74    | 40.5        | 19.2      |
>
>
> However, we must emphasize that **this approach violates the fundamental premise of learning directly from long-tail distributions**. The use of pre-trained balanced teachers essentially bypasses the core challenge of addressing long-tail distributions within adversarial training. For this reason, we deliberately focused on lightweight balanced self-teachers in this paper, as they are explicitly designed to address the unique challenges posed by long-tail datasets. While the use of highly robust teachers remains a viable extension, it diverges from the primary goals of this work and thus was not included in the main experiments.
>
> [1] Chen, Erh-Chung, and Che-Rung Lee. "Ltd: Low temperature distillation for robust adversarial training." arXiv preprint arXiv:2111.02331 (2021).
>
> [2] Wang, Zekai, et al. "Better diffusion models further improve adversarial training." International Conference on Machine Learning. PMLR, 2023.
>
> [3] Cui, Jiequan, et al. "Decoupled kullback-leibler divergence loss." arXiv preprint arXiv:2305.13948 (2023).

---

> ### Author Response · Authors · 2024-11-19
> **Reply to Reviewer 61WA. (3/3)**
>
> **W3. Table 5 is not interpretable.**
>
> **A.**  Sorry for our mistake. We have updated our paper.
>
> **W4. Show exeperimental results when the imbalance ratio is 100.**
>
> **A.**
>
>
>
> The clean accuracy and robustness with 100 of IR using ResNet-18 on CIFAR-10-LT. T-Clean and T-PGD are clean and 20-step of PGD accuracy on the tail class group.
> | **Method**      | **Clean**            | **PGD**              | **AA**               | **T-Clean**         | **T-PGD**          |
> |------------------|----------------------|----------------------|----------------------|---------------------|---------------------|
> | RoBal           | 66.73 ± 0.26        | 22.64 ± 0.28        | 20.34 ± 0.35        | 52.44 ± 2.53       | 6.3 ± 1.09         |
> | Reat            | 64.84 ± 0.52        | 25.93 ± 0.45        | 23.83 ± 0.22        | 50.08 ± 3.43       | 6.3 ± 0.98         |
> | BSL             | 64.05 ± 0.33        | 26.30 ± 0.38        | 24.21 ± 0.34        | 45.02 ± 3.02       | 5.5 ± 1.09         |
> | **Ours**        | **66.10 ± 0.54**    | **30.24 ± 0.34**    | **26.76 ± 0.31**    | **53.04 ± 2.14**   | **12.12 ± 0.97**   |
>
>
>
> The clean accuracy and robustness with 100 of IR using ResNet-18 on CIFAR-100-LT. T-Clean and T-PGD are clean and 20-step of PGD accuracy on the tail class group.
>
> | Method       | Clean             | PGD               | AA                | T-Clean           | T-PGD             |
> |--------------|-------------------|-------------------|-------------------|-------------------|-------------------|
> | RoBal        | 31.22 ± 0.22     | 10.63 ± 0.21     | 9.68 ± 0.17      | 0.28 ± 0.15      | 0.08 ± 0.06      |
> | Reat         | 32.64 ± 0.55     | 10.81 ± 0.24     | 9.81 ± 0.11      | 10.80 ± 0.62     | 1.18 ± 0.08      |
> | BSL          | 33.20 ± 0.34     | 10.66 ± 0.08     | 9.78 ± 0.10      | 10.60 ± 0.60     | 1.22 ± 0.27      |
> | **Ours**     | **34.44 ± 0.21** | **14.63 ± 0.18** | **12.32 ± 0.08** | **11.21 ± 0.61** | **2.60 ± 0.15** |
>
>
> **Q1. Does the re-sampling method matter?**
>
> **A.** Yes, the re-sampling method plays a crucial role in ensuring tail-class robustness. Both our theoretical and experimental analysis emphasize the importance of re-sampling in improving robustness for tail classes. As shown in Figure 5, the self-teacher trained on a balanced dataset contributes to performance improvement, particularly for tail classes robustness.
>
> **Q2. Does the size of the teach model matter?**
>
> **A.** This question slightly diverges from the scope of our experiments, as our method primarily employs a self-teacher rather than relying on external teacher models of varying sizes. The focus is on improving robustness using a balanced self-teacher trained from the same architecture as the student model. However, if you are referring to experiments with different architectures, we provide additional results in Tables 7 and 8
>
>
> **Q3. Does the re-sampled dataset has to be perfectly balanced or it could have some imbalance?**
>
> **A.** Thank you for this very interesting question. While we have not exhaustively experimented with all possible levels of imbalance, we conducted experiments where the self-teacher was trained on a dataset with a reversed long-tail distribution—i.e., with more samples in the tail classes and fewer in the head classes through extensive up-sampling. Interestingly, the tail-class performance in this scenario did not differ significantly from the results obtained using a fully balanced self-teacher. This suggests that while perfect balance is not strictly required, ensuring sufficient representation for the tail classes remains critical for robust performance.

---

> > ### Author Response · Authors · 2024-11-21
> > **Dear reviewer 61WA.**
> >
> > Thank you for reviewing our submission and sharing your valuable feedback.
> >
> > We have carefully addressed your comments and provided detailed responses. We hope our explanations resolve the concerns you raised. If there are any further questions or points requiring clarification, please feel free to let us know. We truly appreciate your time and effort in reviewing our work.
> >
> > Best regards,
> > Authors of paper number 5341.

---

> > > ### Comment · Reviewer_61WA · 2024-11-21
> > >
> > > Thank you for your detailed response. I understand the aspects of novelty you are emphasizing in this paper. However, I still find the technical contributions to the community to be somewhat limited.
> > >
> > > The new results on larger imbalance ratios are promising and look good to me. However, these results represent significant additions that should have been included in the original submission by the deadline. As such, I will not consider them in my final evaluation, as doing so would not be fair to other submissions.
> > >
> > > Overall, my evaluation of the paper remains unchanged.

---

> > > > ### Author Response · Authors · 2024-11-21
> > > > **Reply to Reviewer 61WA.**
> > > >
> > > > Thank you for your thoughtful response and for considering our clarifications.
> > > >
> > > > We understand your stance on excluding additional results added during the rebuttal period and respect your evaluation.
> > > > We appreciate your valuable feedback, which has been instrumental in refining our work.
> > > >
> > > > Thank you again for your time and consideration.

---

### Author Response · Authors · 2024-12-02
**Global Response**

Dear Reviewers,

We sincerely thank you for your insightful questions and constructive feedback. Your comments have been invaluable in improving our work. We have carefully addressed all the raised concerns in our rebuttal and revised manuscript.


First, we deeply appreciate that you recognized the following strengths in our work:
- The theoretical and experimental analysis highlighting how adversarial training reduces robust performance for tail classes, which was noted as an intriguing and valuable contribution.
- The superior performance demonstrated by our method in long-tailed adversarial training scenarios, outperforming existing baselines.

Additionally, we are grateful for the opportunity to clarify and address the important concerns raised in the reviews, including:
- Extending our experiments to evaluate performance under an imbalance ratio (IR) of 100, a setting commonly used in long-tail literature. We found that our method continues to achieve strong performance even in this challenging scenario.
- Revising the Method section to provide a clearer and more intuitive explanation of the integration of balanced softmax and knowledge distillation (KD). Specifically, we expanded the discussion to explain how KD leverages the robustness of a balanced teacher model to improve student performance, particularly for tail classes.
We have also taken steps to enhance the overall clarity of our manuscript and have corrected all identified typos, ensuring the text is as polished and comprehensible as possible.

We sincerely hope these revisions effectively address the reviewers' concerns and contribute to improving the overall quality of our paper.
Thank you once again for your thoughtful review and valuable feedback.

Best regards,

The Authors

---

### Meta-Review · Area_Chair_zRAZ · 2024-12-20

**Metareview:**

This paper presents an in-depth analysis of adversarial robustness in long-tailed datasets and introduces an effective adversarial training approach based on the findings. Its inspiring theoretical analysis and comprehensive experiments are recognized by the reviewers.

Initially, the reviewers raised some concerns. However, most of these issues were successfully addressed during the rebuttal phase, leading all reviewers to favor acceptance of the paper. The recommendation is to accept the paper. The authors should carefully revise their paper by incorporating the reviewers' suggestions.

**Additional Comments On Reviewer Discussion:**

Most of the reviewers' concerns are addressed during rebuttal and all reviewers tend to accept the paper.

---

### Decision · Program_Chairs · 2025-01-22

Accept (Poster)